# Environmental and genetic regulation of *Streptococcus pneumoniae* galactose catabolic pathways

Banaz O. Kareem[1,2,6], Ozcan Gazioglu [1,6], Karina Mueller Brown [3], Medhanie Habtom[1], David G. Glanville [4], Marco R. Oggioni [1,5], Peter W. Andrew[1], Andrew T. Ulijasz [4], N. Luisa Hiller [3] & Hasan Yesilkaya [1]✉

Efficient utilization of nutrients is crucial for microbial survival and virulence. The same nutrient may be utilized by multiple catabolic pathways, indicating that the physical and chemical environments for induction as well as their functional roles may differ. Here, we study the tagatose and Leloir pathways for galactose catabolism of the human pathogen *Streptococcus pneumoniae*. We show that galactose utilization potentiates pneumococcal virulence, the induction of galactose catabolic pathways is influenced differentially by the concentration of galactose and temperature, and sialic acid downregulates galactose catabolism. Furthermore, the genetic regulation and in vivo induction of each pathway differ, and both galactose catabolic pathways can be turned off with a galactose analogue in a substrate-specific manner, indicating that galactose catabolic pathways can be potential drug targets.

Survival and virulence of microbes in their host rely on their ability to efficiently obtain and metabolize the nutrients from the niche they inhabit[1]. For mucosal pathogens, mucin, a high molecular weight glycoprotein found within the composition of mucus, is a nutrient-rich compound[2]. The glycan chains of mucin contain utilizable sugars, including fucose, galactose and galactosamine, glucosamine, mannose, and sialic acid. Among these, galactose is the most abundant sugar, such that up to 45% of mucin sugar content is composed of galactose and galactosamines[3]. Catabolism of galactose occurs either via the tagatose or via the Leloir pathway. Subsequently, the products of these two pathways (tagatose 1,6 bisphosphate and glucose 1-phosphate) enter glycolysis and are oxidized to pyruvate[4]. While both pathways function simultaneously in certain species, for example *Streptococcus lactis*[5] and *Streptococcus mutans*[6], in others either the tagatose, e.g, *Streptococcus cremoris*, or Leloir pathway, e.g., *Streptococcus salivarius*, operate[7]. There are even differences between galactose metabolic pathways in different strains of *Lactococcus lactis* due to the presence or absence of a plasmid encoding tagatose pathway

genes[8]. These pathways have been studied mainly in lactic acid bacteria within the context of industrial microbiology applications, however, their function and regulation have not been studied in detail in an opportunistic human pathogen.

*Streptococcus pneumoniae* is an opportunistic human pathogen and is a causative agent of several important diseases, including otitis media, bacteremia, meningitis, and pneumonia[9]. Conversely, the bacterium also is a frequent occupant of the human nasopharynx, where it resides without causing symptoms. Several of its virulence determinants have been characterized. However, since this pathogen lacks a complete TCA cycle and is therefore reliant on the acquisition and utilization of sugars for host survival[10], the significance of the pneumococcus's ability to acquire and metabolize nutrients within the host and the contribution of these processes to its survival and virulence requires further examination for therapeutic developments. While it can ferment up to 32 different sugars[11], in the respiratory tract the pneumococcus utilizes sugars found within the structure of host glycans, including galactose.

[1]Department of Respiratory Sciences, University of Leicester, Leicester, UK. [2]Department of Basic Medical Sciences, College of Medicine, University of Sulaimani, Sulaimani, Iraq. [3]Department of Biological Sciences, Carnegie Mellon University, Pittsburgh, PA, USA. [4]Department of Microbiology and Immunology, Loyola University Chicago, Maywood, IL, USA. [5]Department of Pharmacy and Biotechnology, Bologna, Italy. [6]These authors contributed equally: Banaz O. Kareem, Ozcan Gazioglu. ✉e-mail: hy3@le.ac.uk

It has been established that galactose has a significant impact on pneumococcal survival and virulence[12,13]. The catabolism of galactose is carried out by two galactosidases, BgaA and BgaC, which are specific for Galβ1-3GlcNAc and Galβ1-4GlcNAc linkages, respectively[14], and are highly upregulated in the presence of mucin (but not glucose)[14]. In addition, genes potentially involved in the transport and intracellular metabolism of galactose result in a high level of induction under the same conditions. Taken together, the presence of multiple galactosidases and transporters and their respective regulation in the presence of mucin suggests that the pneumococcus has evolved to release galactose from different host substrates.

We have demonstrated that the pneumococcus can grow on galactose and that both BgaA and BgaC galactose catabolic pathways are operative in *S. pneumoniae* cells growing on galactose as the sole carbon source[14]. Interestingly, homologs for all the genes in the Leloir and tagatose pathways have been found within the genome sequences of *S. pneumoniae*. The importance of galactose metabolism for pneumococcal virulence was reinforced after analysis of the consequences of disruption of the two pathways by mutation[14]. Our published data show that, unlike the mutation of genes of N-acetyl glucosamine and mannose metabolism, which did not have a significant impact on virulence, the mutation of a key gene of the Leloir pathway (*galK*), or the tagatose pathway (*lacD*), or in combination, reduces pneumococcal colonization and virulence[14].

Galactose utilization has also been linked to biofilm formation, increased biomass, and neuraminidase production in *S. pneumoniae*[12–14]. Importantly, the synthesis of the pneumococcal polysaccharide capsule, which is essential for virulence, is affected by galactose utilization in a type 2 D39 strain. Some of the capsule precursors are generated through the catabolism of galactose through the Leloir pathway such as α-glucose-1-phosphate (α-G1P). Critically, although not surprisingly, it has been shown that sugar catabolism is a determinant of the extent of pneumococcal encapsulation in a type 2 D39 strain, which produced a larger quantity of capsules when grown on galactose than on glucose[15]. Thus, galactose metabolism shapes transcription, population-level behaviors, and virulence determinants.

Some pertinent questions that arise are: What specific function(s) does each pathway play during infection of different host tissues? What are the genetic and environmental determinants of the activities of galactose catabolic pathways? In this study, we began by investigating the environmental determinants of pneumococcal galactose metabolism using in vitro conditions mimicking host airways, such as different sugars and oxygen concentrations, and at different temperatures. We then determined the genetic basis of the regulation of the individual pathways. Finally, we identified when each pathway is induced after intranasal infection and studied the role of each pathway in the expression of selected genes in vivo.

## Results

### Growth on galactose enhances pneumococcal virulence

To determine the impact of different sugars on pneumococcal virulence, we prepared an infective dose using pneumococcal cultures grown in a chemically defined medium (CDM) supplemented either with 55 mM glucose or galactose (Fig. 1). Mice were infected intranasally or intravenously and the survival time was recorded in a week-long assay. Using the intranasal route, 4 out of 5 mice survived with the dose prepared on glucose, while, with galactose 1 out of 5 mice survived ($p < 0.01$). These results suggest that bacteria in the lungs are more virulent when they have the opportunity to initially metabolized galactose rather than glucose as the sole carbon source.

In contrast, following the intravenous route of infection, we found that the virulence potential was independent of the sugar source. All mice needed to be culled by 43.5 h (SD: 12.4) or 52 h (SD:12.6) for the cohort that received the dose propagated on glucose or galactose, respectively ($p > 0.05$). These results show that the metabolic profile of the bacteria before infection can influence virulence and that this effect is site-specific. Particularly, a metabolic state primed to metabolize galactose increases virulence in the respiratory tract. However, the high pathogenic potential of the pneumococcus in the blood appears to be independent of the metabolic state of the bacteria (at least when considering glucose versus galactose) at the time of infection.

The pneumococcal cultures grown on galactose are known to produce more capsules than those grown on glucose[15]. Therefore, to determine whether the potency of virulence associated with the galactose-grown cultures is linked to a higher level of capsule synthesis, we evaluated the capsule level of the pneumococci on glucose versus galactose growth. The results showed that on galactose ($10.5 \pm 0.2 \, \mu M/10^8$ CFU), the type 2 D39 strain synthesized more capsules than on glucose ($3.8 \pm 0.3 \, \mu M /10^8$ CFU). This shows that, in addition to the metabolic adaptation that allows efficient utilization of galactose, a higher capsule level could also be responsible for potentiating the impact of galactose growth.

### Environmental regulation of galactose catabolic pathways

The results presented in Fig. 1, clearly established that the utilization of galactose is important for pneumococcal virulence in the respiratory tract. Further, our previous work[14] established that each galactose catabolic pathway contributes to pneumococcal virulence in the respiratory tract. However, the extent to which each pathway is active in any given condition remains unknown.

The pneumococcus is exposed to different environments in vivo during infection. For example, the upper respiratory tract has a high concentration of oxygen (20%) and a relatively low temperature, 34 °C, whereas the lower respiratory tract is nearly anaerobic and higher temperature, 37 °C[16,17]. In addition, sugar composition and quantity on mucosal surfaces will vary depending on the distribution of mucin-producing cells, and glycosylation pattern of mucins[18]. To determine the effect of changing levels of temperature, oxygen concentrations, and different nutrients on the catabolic pathways, we used *lacZ* reporter strains to measure the induction of either *lacA* or *galK*, the key genes of the tagatose and Leloir pathways, respectively.

**Nutrients.** Our results showed that while the tagatose pathway was induced with as low as 1 mM galactose relative to without galactose (Fig. 2a), the Leloir pathway required 20 mM galactose for a significant induction (Fig. 2b). Moreover, we consistently obtained 5 to 6-fold higher Miller units for the same number of cells with P*lacA* than with P*galK*. In the presence of 1- or 2.5 mM glucose, the induction of both

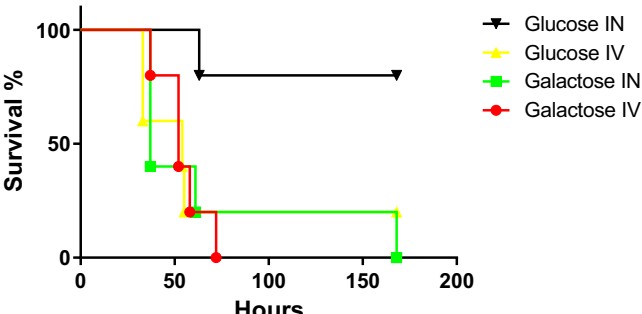

**Fig. 1 | The impact of glucose and galactose on pneumococcal virulence after intranasal (IN) and intravenous (IV) infection.** S. *pneumoniae* D39 strain was propagated in CDM containing either 55 mM glucose or galactose. A group of 5 female MF1 mice received the dose either intranasally (Log10 CFU/ml: for galactose 6.33 for glucose 6.47/mouse) or intravenously through the tail vein (Log10 CFU/ml: for galactose 5.84, for glucose 6.12/mouse). Mice were observed for disease signs over 1 week. The percent survival was analyzed using the Mantel-Cox test. Source data are provided as a Source Data file.

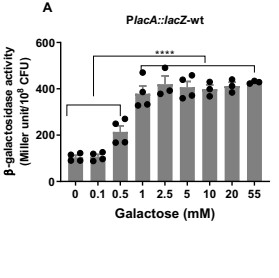
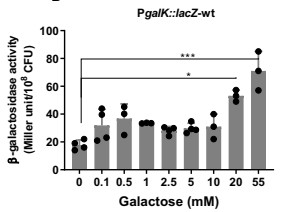
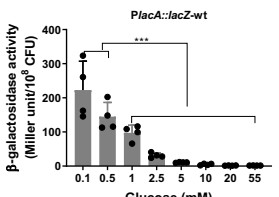
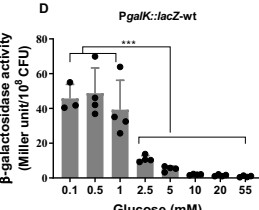

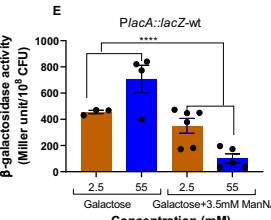
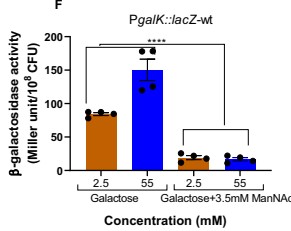
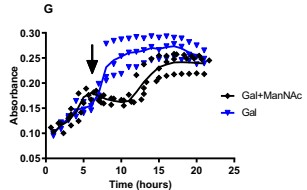

**Fig. 2 | The expression of galactose catabolic pathways is sugar dependent and ManNAc represses both pathways in galactose.** Inductions of P*lacA::lacZ*-wt (**A** and **C**) and P*galK::lacZ*-wt (**B** and **D**) were analyzed 4 h after induction in CDM supplemented with different galactose (**A** and **B**) and glucose concentrations (**C** and **D**) ('P' stands for promoter). The addition of 3.5 mM ManNAc represses the induction of galactose-induced P*lacA* (**E**) and P*galK* (**F**) regardless of galactose concentration, and stalls pneumococcal growth (**G**). The downward arrow shows when ManNAc was added. The mean of at least three biological replicates with their SEM is given. Data was analyzed by ordinary one-way ANOVA followed by Tukey's multiple comparisons test. \**p* = 0.03, \*\*\**p* = 0.0001, \*\*\*\**p* < 0.0001. Source data are provided as a Source Data file.

pathways decreased significantly compared to 0.1 mM, and the repression increased with elevated concentrations of glucose (Fig. 2c and d). Thus, we conclude that the tagatose pathway is more sensitive to low galactose concentrations than the Leloir pathway and that both pathways are similarly repressed by glucose.

An additional relevant environmental determinant of galactose utilization is the presence of sialic acid, generated by sialidase (NanA) activity[19]. Galactose and sialic acid are the most accessible sugars following the sequential deglycosylation of mucin polysaccharides[19]. The utilization of sialic acid follows a well-conserved path in bacteria[20]. After transport into the cell, sialic acid is dissimilated into N-acetylmannosamine (ManNAc) and pyruvate. While pyruvate is utilized via the oxidative or fermentative routes, ManNAc is phosphorylated by NanK to form ManNAc-6-P, which is then utilized as a substrate by epimerase (NanE) to generate GlcNAc-6-P. Subsequently, GlcNAc-6-P is deacetylated and deaminated to form fructose-6-P via the actions of NagA and NagB, respectively. The pneumococcal D39 serotype 2 strain is unable to catabolize sialic acid due to mutations in the neuraminate lyase gene[20]. Hence, to study the impact of sialic acid on galactose catabolic pathways, we used the utilizable analog of sialic acid, ManNAc. ManNAc had a profound effect on the induction of P*lacA* and P*galK* irrespective of galactose concentration, and growth on galactose, consistent with a switch to preferentially process ManNAc (Fig. 2e, f). To investigate this from another perspective, we measured gene expression after the addition of ManNAc. RNAseq data show that within 15 mins of the addition of ManNAc to the pneumococcal culture growing in the presence of galactose, there is down-regulation of genes coding for the four galactose uptake systems and the two galactose catabolic pathways relative to galactose alone (Table 1). Together, these findings suggest that after the addition of ManNAc there is a substrate-induced metabolic switch to firstly catabolize the ManNAc rather than the galactose (Fig. 2g).

To determine whether we could reproduce our results with a strain that has an intact neuraminate lyase gene, we used the serotype 4 TIGR4 strain. When we added 3.5 mM sialic acid to galactose-grown pneumococci, a substrate-induced metabolic switch could also be demonstrated with the TIGR4 strain by growth studies (Supplementary Fig. 1). Moreover, the addition of either 3.5 mM ManNAc or sialic acid to the pneumococcal cultures grown on galactose led to a significant decrease in the expression of *galK* (3.4 ± 0.07 and 6.6 ± 0.03, *n* = 4 for ManNAc and sialic acid, respectively) and *lacD* (4.5 ± 0.15 and 2.8 ± 0.25, *n* = 4 for ManNAc and sialic acid, respectively) compared to the growth on galactose. Therefore, our observation that sialic acid addition decreases the expression of both galactose catabolic pathways and leads to a substrate-induced metabolic switch is reproducible and not strain-specific.

**Temperature and oxygen concentration.** Next, we tested the impact of temperature and oxygen concentration on the promoter induction of *lacA* and *galK* in the presence of 55 mM galactose or glucose; we selected this concentration as both pathways are highly induced in the presence of galactose and highly repressed in glucose. On galactose, tagatose pathway induction decreased significantly when the temperature increased to 39 °C compared to 33 °C and 37 °C (*p* < 0.001) (Fig. 3a). In contrast, the temperature had no significant impact on the P*galK* induction (*p* > 0.05) (Fig. 3b). We conclude that temperature influences the tagatose pathway and that it may be repressed under febrile conditions.

The effect of oxygen on both pathways was determined by growing the reporter strains in anaerobic and microaerobic conditions. On galactose, both catabolic pathways had reduced induction without oxygen (*p* < 0.001). On glucose, while P*lacA* was induced anaerobically, no change was observed in the induction of P*galK* regardless of oxygen (Supplementary Fig. 2). Together, these results show that temperature and oxygen can impact the induction and repression of galactose catabolic pathways. Further, the effects differ between the pathways, allowing for changes in the relative contributions of each pathway in diverse conditions.

In summary, we have measured the response of the tagatose and Leloir pathways to multiple metabolites (galactose, glucose, and sialic acid derivative), temperature, and oxygen. Both pathways are induced by galactose, and repressed by glucose and sialic acid, suggesting the latter is preferentially metabolized over galactose. Moreover, the tagatose pathway is turned on at lower concentrations of galactose, which is consistent with this pathway taking a major role in these conditions. In addition, while the tagatose pathway is sensitive to

**Table 1 | The expression of selected genes**

| Gene name | Gene number | Function | GAL+ManNAc +15 versus GAL | GAL+ManNAc + 30 versus GAL | ManNAc versus GAL |
|---|---|---|---|---|---|
| | SPD_0069 | PTS system transporter subunit IIA | 0.20 | 0.33 | 0.56 |
| | SPD_0088 | ABC transporter permease | 0.41 | 1.65 | 3.24 |
| | SPD_0090 | ABC transporter substrate-binding protein | 0.27 | 1.37 | 4.00 |
| | SPD_0262 | PTS system, mannose/fructose/sorbose family protein, IID component | 0.37 | 0.60 | 2.07 |
| manM | SPD_0263 | PTS system mannose-specific transporter subunit IIC | 0.49 | 0.71 | 2.29 |
| manL | SPD_0264 | PTS system mannose-specific transporter subunit IIAB | 0.53 | 1.11 | 3.78 |
| | SPD_0560 | PTS system transporter subunit IIB | 0.19 | 0.34 | 0.42 |
| | SPD_0561 | PTS system transporter subunit IIC | 0.14 | 0.30 | 0.37 |
| lacD | SPD_1050 | tagatose 1,6-diphosphate aldolase | 0.10 | 0.19 | 0.28 |
| lacC | SPD_1051 | tagatose-6-phosphate kinase | 0.10 | 0.25 | 0.36 |
| lacB | SPD_1052 | galactose-6-phosphate isomerase subunit LacB | 0.13 | 0.22 | 0.38 |
| lacA | SPD_1053 | galactose-6-phosphate isomerase subunit LacA | 0.11 | 0.26 | 0.41 |
| - | SPD_1632 | hypothetical protein | 0.25 | 0.41 | 0.13 |
| galT-2 | SPD_1633 | galactose-1-phosphate uridylyltransferase | 0.24 | 0.43 | 0.10 |
| galK | SPD_1634 | galactokinase | 0.27 | 0.49 | 0.13 |
| galR | SPD_1635 | galactose operon repressor | 0.20 | 0.32 | 0.15 |

The fold change in the presence of galactose (GAL) and N-acetyl mannosamine (ManAc) after 15- and 30-min incubation, and the fold change in ManAc relative to GAL are shown.

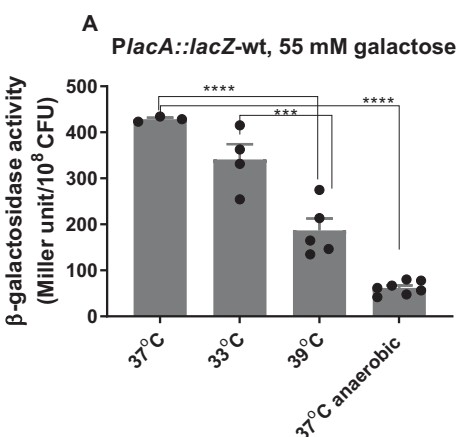
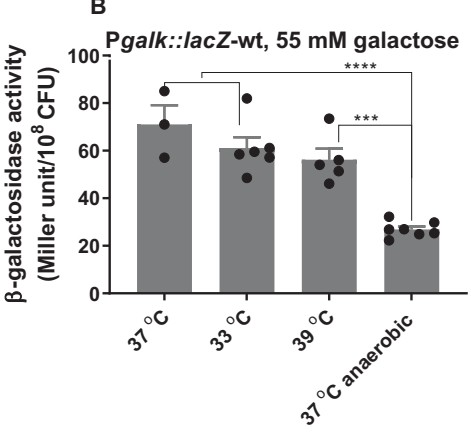

**Fig. 3 | Impact of environmental factors on the expression of galactose catabolic pathways.** Analysis of P*lacA* (**A**) and P*galK* (**B**) induction in CDM supplemented with 55 mM galactose incubated 4 hours microaerobically at different temperature or anaerobically at 37 °C using LacZ reporter assay. Statistical analysis was performed using one-way ANOVA followed by Tukey's multiple comparisons test, ***$p$ = 0.0005, ****$p$ < 0.0001. The mean of at least three biological replicates with their SEM is given. Source data are provided as a Source Data file.

temperature changes between 33 °C and 39 °C, the Leloir pathway is not, suggesting that febrile conditions change the relative balance between the pathways and thus the byproducts of galactose metabolism.

### Galactose catabolic pathways influence biofilm development

The pneumococcus forms biofilms during chronic infections[21], thus we thought to test whether and how galactose catabolism pathways influence biofilm development. To this end, we used crystal violet assays to compare biofilm formation for a WT strain, as well as deletions mutants of *lacA* (Δ*lacA*), galK (Δ*galK*), and both (Δ*lacA*Δ*galK*). Strains were grown for 24 h in BHI, a condition where strains have comparable growth (Fig. 4c). The WT and Δ*galK* displayed the same levels of biofilm, while the Δ*lacA* displayed a trend toward more biofilm, however these data were not statistically significant. In contrast,

the double mutant (Δ*lacA*Δ*galK*) displayed a significant increase in biofilm mass (Fig. 4a, b). In different conditions, biofilm formation has been observed to be higher in galactose than glucose[12]. These data show that galactose catabolism through both pathways can decrease biofilm formation in rich media conditions. Although these conditions cannot perfectly mimic that of host colonization, we can conclude that the processing of carbohydrates plays a key role in the regulation of biofilm development.

### In vivo expression and regulation of galactose catabolic pathways

We previously showed that both the Leloir and tagatose pathways play a significant role in pneumococcal colonization and virulence[14]. However, it remains enigmatic as to which stage(s) of infection the individual pathways are active. We hypothesized that galactose catabolic

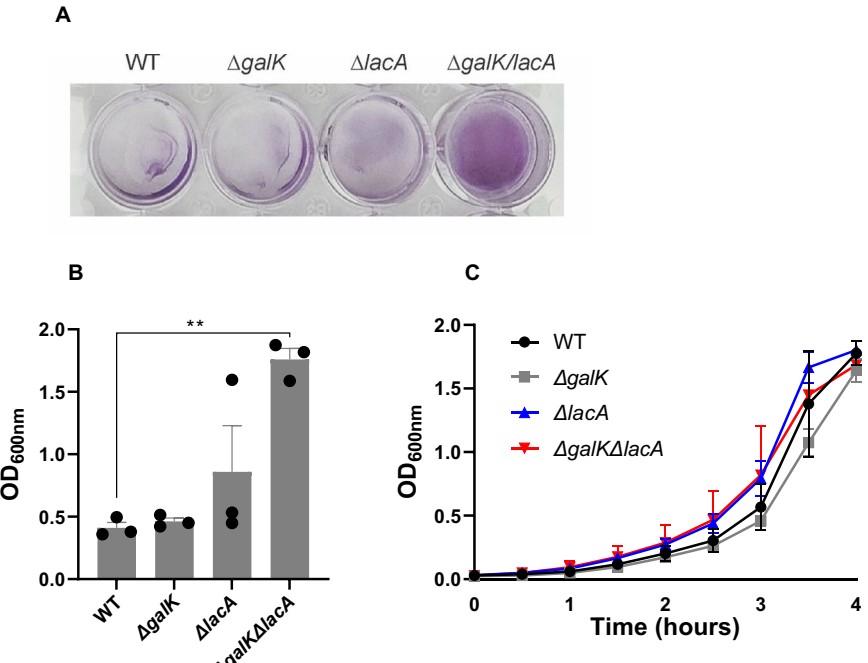

**Fig. 4 | Galactose catabolism pathways influence biofilm formation in vitro.**
**A** Strains were seeded at low optical density and grown in BHI at 37 °C for 24 h and then stained with crystal violet (representative image). **B** Quantification of crystal violet staining for WT and deletions mutants of *lacA* (Δ*lacA*), galK (Δ*galK*), and both (Δ*galK/lacA*). Vertical lines show SEM. **C** Growth curve of WT and mutant strains in BHI at 37 °C. The mean of at least three biological replicates with their SEM is given. Statistical analysis was performed using One-way ANOVA followed by Tukey's multiple comparisons test (**$p = 0.0047$). Source data are provided as a Source Data file.

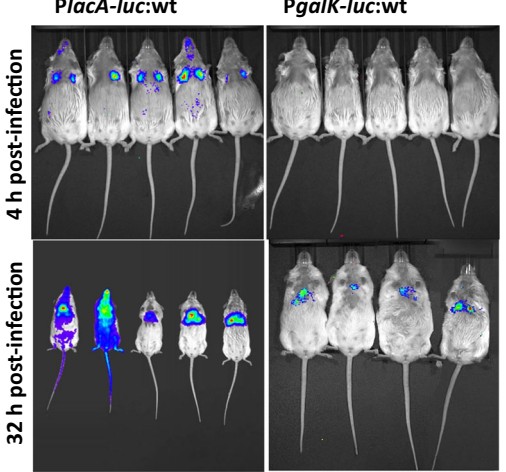

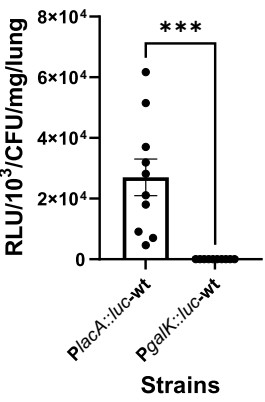

**Fig. 5 | Temporal expression of galactose catabolic pathways in a murine model of *S. pneumoniae*.** Mice were infected intranasally approximately with $5 \times 10^6$ CFU/mouse either with P*lacA-luc*:wt or P*galK-luc*:wt. Animals were scanned either at 4 h or 32 h post-infection. Bioluminescence was quantified after subcutaneous luciferin injection. Experiments were done twice using a total of 10 mice for each group. Here representative images are given. The graph depicts the relative light unit (RLU) normalized against the colony-forming units for readings at 4 h. Data represent the mean RLU derived from 10 mice with SEM and were analyzed using a two-sided *t* test, ***$p = 0.0003$. Source data are provided as a Source Data file.

pathways would be expressed at different times after pneumococcal infection. To measure the spatiotemporal pattern of induction, mice were infected with pneumococcal strains carrying either P*lacA::luc*, *lacA* is the first gene of the Lac operon, or P*galK::luc* in a single copy. Using a model of pneumonia, mice were infected intranasally and promoter activity was measured at 4- and 32 h post-infection by scanning for bioluminescence after administering luciferin. Strikingly, we found that P*lacA* was induced by 4 h post-infection, however, no induction of P*galK* could be seen at this early time point (Fig. 5). At 32 h post-infection, bioluminescence from P*lacA* progressively increased and reached $3.53 \times 10^6$ (SEM: $7.06 \times 10^5$) RLU, and a signal for P*galK* also could also be detected, albeit significantly lower than P*lacA* ($5.38 \times 10^5$, SEM: $1.29 \times 10^5$) ($p < 0.01$) (Fig. 5). These data suggest that the tagatose pathway is active in the early phase of infection, while the Leloir pathway is active at a later stage of infection.

We also hypothesized that the importance of galactose catabolic pathways in pneumococcal virulence and colonization could be either due to their effect on capsule biosynthesis, or alternatively on cell-cell

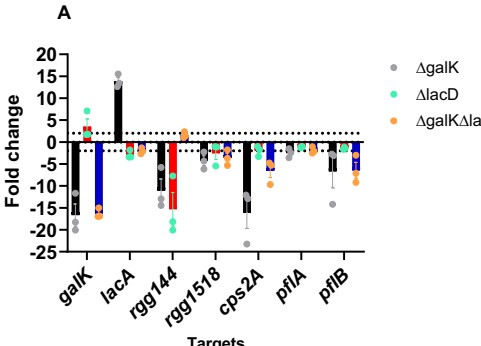

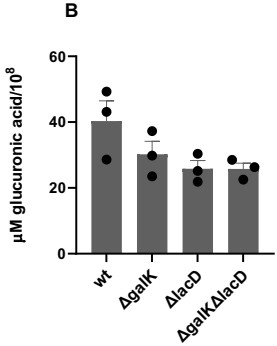

**Fig. 6 | Influence of galactose catabolism on gene expression and capsule.** Expression of selected genes (**A**) and quantification of glucuronic acid in galactose catabolic pathway mutants relative to WT. Values above and below (for downregulation) the dotted lines represent the significant levels of expression (**B**). Gene expression and glucuronic acid levels were quantified in pneumococci obtained 4 h after intranasal infection through lung lavage. Each column in (**A**) and (**B**) represents the mean of three experiments. For each experiment lung lavage from two mice were used. The mean recovered CFU for different strains was similar: D39: 1.81X10[7], ΔgalK: 1.9X10[7], ΔlacD: 1.7X10[7], and ΔgalKΔlacD: 1.76X10[7]. Vertical lines show standard deviation. Source data are provided as a Source Data file.

communication systems. Capsule biosynthesis is influenced by galactose utilization as some of the precursors of capsule biosynthesis are generated through galactose catabolism[22]. Furthermore, our previous work showed that the Rgg/SHP144 and Rgg/SHP1518 cell-cell communication systems are responsive to galactose[23–26]

To understand how the catabolic pathways contributed to pneumococcal virulence, we evaluated the expression of selected genes in *ΔgalK*, *ΔlacD*, and *ΔgalKΔlacD* recovered 4 h post-infection from the lungs (Fig. 6a). The results showed a significant induction of *galK* in *ΔlacD* and *lacA* induction in *ΔgalK*. This suggests that the lack of one pathway can be compensated by the other in vivo. As expected, both *galK* and *lacA* were significantly downregulated in *ΔgalKΔlacD* ($p < 0.001$). Expression of the *cps2A* capsule locus was downregulated in all three mutants, but the highest reduction was seen in *ΔgalK*, consistent with the involvement of the Leloir pathway in generating α-glucose-1 phosphate as a precursor for capsule biosynthesis[15]. In addition, we detected significant downregulation in *ΔgalK*, but not in *ΔlacD*, of genes coding for pyruvate formate lyase (*pflB*) and pyruvate formate lyase-activating enzyme (*pflA*) ($p < 0.001$), both of which are activated in the presence of galactose and are linked to mixed acid fermentation[27]. Decreased expression of *pflA* and *pflB* is consistent with the attenuated galactose metabolism, hence its impact on the decreased synthesis of pyruvate, which is used as a substrate for mixed acid fermentation. It should be noted that PflB requires post-translational activation by PflA. Hence, mRNA levels might not be reflective of the actual PflB activity. We also determined the expression of cell-cell communication systems, *rgg144* and *rgg1518*, shown to be involved in galactose metabolism[23–26,28]. In the absence of these systems, the pneumococcus is attenuated in colonization and virulence very likely due to their effect on galactose utilization and capsule synthesis[23–26,28]. We hypothesized that their impact on galactose utilization may be due to their regulatory effect on galactose catabolic pathways. We found that both *rgg144* and *rgg1518* were downregulated in *ΔgalK*, but only *rgg144* was significantly downregulated in *ΔlacD*. On the other hand, in *ΔgalKΔlacD* only *rgg1518* was significantly downregulated ($p < 0.001$), which we ascribe to the absence of galactose or its intermediates in the double mutant.

Having seen the downregulation of *cps2A* in *ΔgalK*, *ΔlacD* and *ΔgalKΔlacD*, we tested the impact of individual galactose catabolic pathways singly and in combination on induction of the capsule locus in vivo by assaying for glucuronic acid levels because it is found in the type 2 D39 capsule. To do this, mice were infected intranasally and lung aspirates were collected 4 h post-infection. Before infection, no difference in capsule levels was seen among the pneumococcal strains propagated on BHI: wild type ($64 \pm 0.3\,\mu M/108\,CFU$, $n = 4$), *ΔgalK*

($63 \pm 0.5\,\mu M/108\,CFU$, $n = 4$), *ΔlacD* ($58 \pm 0.5\,\mu M/108\,CFU$, $n = 4$) and *ΔgalKΔlacD* ($57 \pm 0.2\,\mu M/108\,CFU$, $n = 4$) ($p > 0.05$). As an assay control, we also included *Δcps* ($39 \pm 0.1\,\mu M/108\,CFU$, $n = 4$), which had significantly less capsule synthesis compared to the wild type ($p < 0.05$). As shown in Fig. 6b, while the glucuronic acid level appeared to be lower in *ΔgalK*, *ΔlacD*, and *ΔgalKΔlacD* than in the wild type, the difference was not statistically significant ($p > 0.05$). The difference in gene expression of *cps2A* and capsule levels in *ΔgalK*, *ΔlacD*, and *ΔgalKΔlacD* may have occurred due to the differences in the length of time required for both transcription as well as posttranslational control[28,29].

## Genetic regulation of galactose catabolic pathways

The data above clearly showed the importance of the two catabolic pathways on *rgg144* and *rgg1518*, supporting their responsiveness to galactose (Fig. 6a). We then tested if these cell-cell communication systems had an impact on the Leloir and tagatose pathways. To do this, we evaluated the induction of P*lacA* and P*galK* in both wild-type and mutant backgrounds (Fig. 7a, b). Results showed that P*lacA* induction was abolished in *Δrgg144* and significantly diminished in *Δrgg1518* ($p < 0.0001$), while P*galK* was induced in *Δrgg144* ($p < 0.001$).

Next, we tested whether these Rgg cell-cell communication systems would control the induction of P*lacA* in vivo, as observed in vitro. We selected P*lacA* because it was induced at an early stage of infection and influenced by both Rgg144 and Rgg1518. For this experiment, we inserted the P*lacA::luc* reporter into *Δrgg144* and *Δrgg1518* strains at an autonomous region of the chromosome where its activity could be monitored without disruption of gene function. The mice were infected intranasally with P*lacA::luc* reporters in the wild type and the mutant backgrounds and scanned four hours post-infection (when there was no significant difference in the colony counts of bacterial strains in the lung tissues) (Fig. 7c, d). Results showed that the expression of P*lacA* decreased significantly in *Δrgg1518* ($p < 0.05$), while deletion of *rgg144* had no significant impact on P*lacA* induction at ($p > 0.05$). Thus, both in vitro and in vivo data suggest that Rgg1518 regulation promotes the tagatose pathway. In contrast, while Rgg144 dramatically promotes the tagatose pathway in vitro, we did not see the same result in vivo, suggesting there may be an alternative pathway that can overcome Rgg144 regulation of the tagatose pathway in vivo, at least early during infection.

## The effect of galactose analogs on pneumococcal growth

To determine if galactose metabolism could be a viable target to control pneumococcal growth, we tested the impact of the galactose analog, 3-Deoxy-3-fluoro-D-galactose (Gal-3F), on pneumococcal

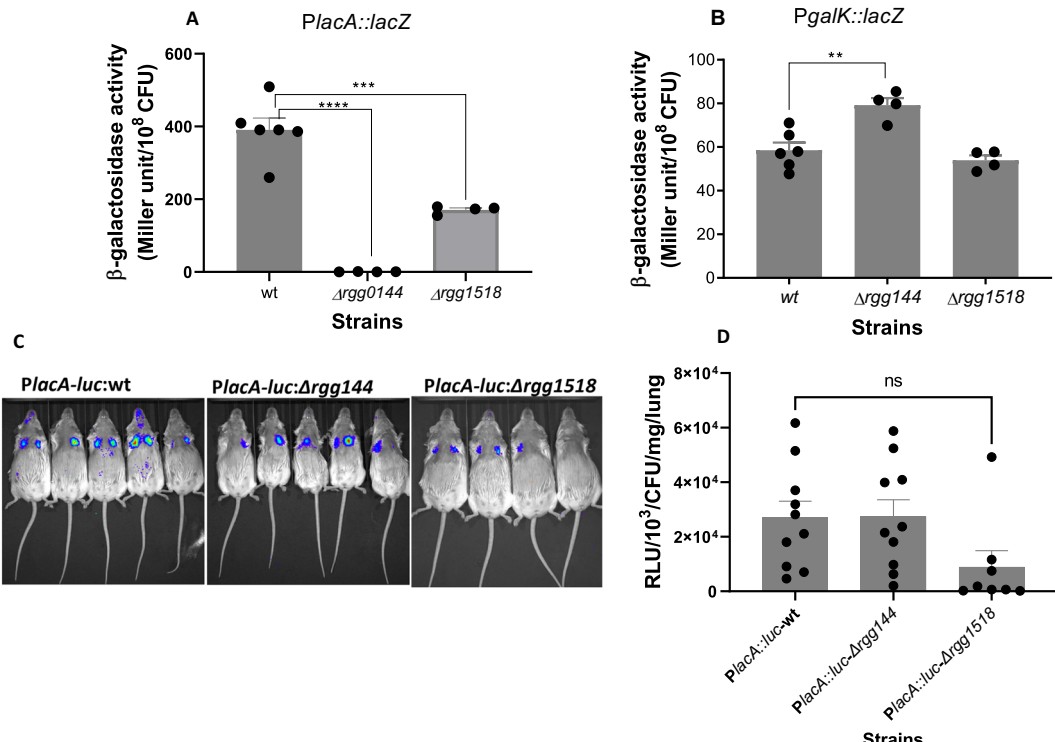

**Fig. 7 | Genetic regulation of tagatose and the Leloir pathways.** Induction of (**A**) P*lacA* and (**B**) P*galK* in different cell-cell communication system mutants. Strains were grown in CDM supplemented with 55 mM galactose. Miller units were normalized against the colony-forming units. Mean data represents the mean of at least 3 biological replicates in duplicate. Vertical lines show the standard error of the mean. ***p* = 0.002, ****p* = 0.0002, *****p* < 0.0001 (**C**) Quantification of the tagatose pathway in a murine model of pneumonia by in vivo imaging. Mice were infected intranasally approximately with 5 × 10⁶ CFU/mouse either with P*lacA-luc*:wt or

P*lacA-luc*:Δrgg144 or P*lacA-luc*:Δrgg1518. Animals were scanned at 4 h post-infection, when there was no difference in the recovered colony counts among the strains, P*lacA-luc*:wt: 4.45 × 10⁶, P*lacA-luc*:Δrgg144: 4.75 × 10⁶ and P*lacA-luc*:Δrgg1518: 4.3 × 10⁶. Experiments were done twice using a total of 10 mice for each group. Here representative images are given. **D** Bioluminescence was quantified after subcutaneous luciferin injection (**p* = 0.036). Data in (**A**), (**B**), and (**D**) were analyzed by ordinary one-way ANOVA followed by Tukey's multiple comparisons test. Source data are provided as a Source Data file.

growth. When pneumococci were grown in CDM supplemented with glucose, the analog had no impact on the growth at any concentration tested (Fig. 8a). However, when the analog was tested in CDM supplemented with galactose, the growth was inhibited at the lowest tested concentration, 17 μM (Fig. 8b). In addition, Gal-3F reduced galactose induction of P*lacA::luc* in a concentration-dependent manner, whereas P*galK::luc* induction was totally abolished with both 10- and 15 μM Gal-3F (Fig. 8c). These results suggest that this analog could be an inhibitor of the pneumococcus in conditions where galactose in the main source of carbohydrates (such as during carriage).

## Discussion

We have a long-term hypothesis that galactose metabolism is a critical factor in pneumococcal survival in vivo[14,23,24,26,27,30–32]. Previously, we demonstrated the importance of both galactose catabolic pathways in colonization and virulence[14], and here we showed that pneumococci propagated on galactose are more virulent than those grown on glucose within the respiratory infection route. In this study, we studied how the environment controls the relative balance between the pathways.

We propose that the galactose-propagated pneumococci are more virulent in vivo relative to those propagated in glucose because they are already metabolically adapted to the environment of the respiratory tract, and they produce more capsules. Galactose is abundant in the respiratory tract, hence, after in vitro growth on galactose, when in the respiratory tract the pneumococcus does not need pre-habituation unlike the microbe grown in glucose, which is found around 0.1 mM in respiratory tract secretions[33]. In addition to supporting growth, galactose pathways can directly potentiate

pneumococcal virulence through their impact on capsule biosynthesis. Supporting this conclusion, our data show downregulation of capsule locus expression in Δ*galK*, Δ*lacD*, and Δ*galK*Δ*lacD* mutants recovered from the lung aspirates, relative to the wild type. In addition, in Δ*galK* and Δ*lacD*, *rgg144* is downregulated. Rgg144 is known to be induced by galactose and represses capsule locus expression[26]. In the absence of Rgg144 repression, capsule locus expression was expected to be higher. Therefore, this adds to our existing knowledge that the regulation of capsule locus expression is multifactorial[34].

Capsule biosynthesis is a highly complex process in *S. pneumoniae* and can be influenced by the capsule type, promoter sequence (regulation), and carbon source[34–36]. Troxler et al. (2019) showed that serotype 7 F pneumococci synthesized significantly fewer capsules than other tested serotypes (6B, 6 C, 9 V, 15, and 23 F) when cultured on glucose or sucrose[36]. A recent study showed that the TIGR4 strain produces more capsules when cultivated on 1 g/L glucose and 0.6 g/L sialic acid than on 0.6 g/L galactose and 0.3 g/L sialic acid[35]. While differences in capsule level in TIGR4 on glucose and galactose can be due to the role of these sugars on capsule biosynthesis in this genetic background, it can be also due to sialic acid's suppressive effect on galactose metabolism as we demonstrated in this study, or the differences in biomass between the two culture conditions[14].

The absence of *galK* and *lacD* decreased the expression of *rgg144* RNA, but *rgg1518* RNA expression decreased only in the absence of the tagatose pathway. Differential expression of *rgg144* and *rgg1518* in the galactose catabolic pathway mutants may show that *rgg144* and *rgg1518* are induced through the metabolites generated by the galactose catabolic pathways. Moreover, distinct expression patterns show that despite their conserved structures[28,37], the induction of each *rgg*

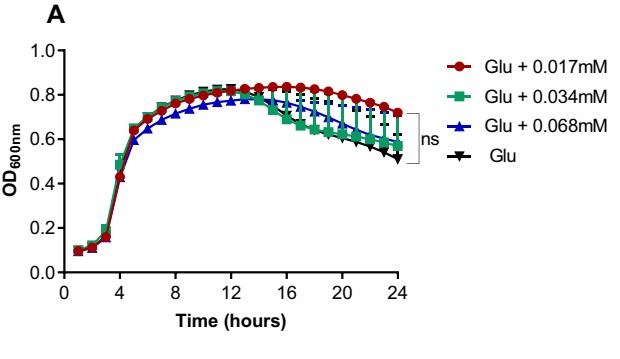

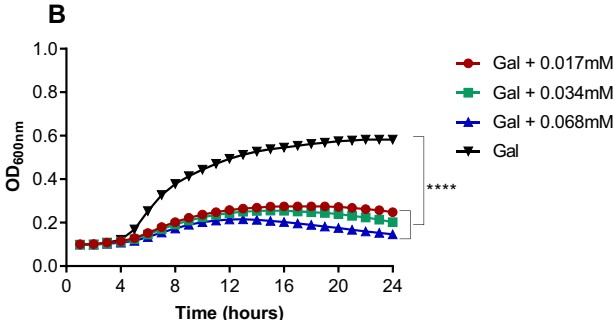

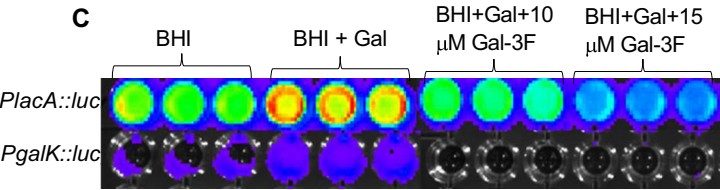

**Fig. 8 | A galactose analog inhibits pneumococcal growth when galactose is the carbohydrate source.** We tested the impact of the Gal-3F analog on pneumococcal growth. Pneumococci were grown either in the presence of glucose (**A**) or galactose (**B**) with different concentrations of Gal-3F. The mean of at least three biological replicates with their SEM is given. Statistical analysis was performed using One-way ANOVA followed by Tukey's multiple comparisons test, 'ns' not significant, ****$p < 0.0001$. **C** Gal3-F impact on the induction of *PlacA::luc* and *PgalK::luc* in different culture media with or without galactose (Gal). The least to the highest induction was represented with colors in the following order: purple, blue, green, and yellow/orange. Source data are provided as a Source Data file.

requires a specific stimulus, hence their unique role for the pneumococcus niche-based growth and invasiveness. Given that *rgg144* is part of the core genome and *rgg1518* is found only in 38% of analyzed pneumococcal strains[26], we speculate that the Leloir pathway emerged before the tagatose pathway in the evolutionary time frame. When the induction of *PlacA* and *PgalK* were tested in *Δrgg144* and *Δrgg1518* backgrounds in vivo, it was found that *rgg144* and *rgg1518* induced the tagatose pathway, but *rgg144* decreases the Leloir pathway. RT-PCR and IVIS results show that there is no bidirectional relationship between *rgg144* and the Leloir pathway expression, probably because of the complexity of regulation in vivo.

Tests of reporter strains indicated that tagatose pathway requires as low as 1 mM galactose for induction, while significant induction of the Leloir pathway needed 20 mM. The early in vivo induction of the tagatose pathway was also supported by IVIS studies in vivo. Differences in the requirement of galactose concentration and the timing of induction of each pathway in vivo may be related to the efficiency of galactose transporters feeding into each pathway. By mutant analysis, we linked four transporters to galactose transport, one ABC, and three PTS transporters[11]. Catabolism via the Leloir pathway is generally linked to the uptake of galactose via a non-PTS permease (secondary carriers or ABC transporters)[38]. ABC transporters are known to impose significant costs because transport can consume up to 60% of cellular ATP[38]. For example, sugar transport via ABC transport requires 1 or 2 ATPs and an additional ATP for phosphorylation, while transport and phosphorylation are coupled in uptake via PTS, and only a single ATP is required[39]. Hence, if the Leloir pathway is served by an ABC transporter and tagatose pathway by the PTS transporters, these significant differences could affect the concentration needed and the timing of the pathway induction.

Several other parameters also have an impact on the induction of the two individual catabolic pathways. High temperature induces the tagatose pathway whereas the Leloir pathway is not affected, implying that the tagatose pathway may be active during the acute stage and inducible, while the Leloir pathway is expressed constitutively

irrespective of temperature fluctuations. This may suggest differences in the stability and activity of enzymes of each pathway at different temperatures. In addition, both pathways are induced in microaerobic relative to anaerobic conditions. The likely reason for this is the positioning of mucin, the source of galactose, in the apical epithelial surface, where the oxygen concentration is higher than the basal membrane section of the respiratory tract. This shows that the pneumococcus has adapted a metabolic machinery that is linked to the environment of substrate that it utilizes.

Our data show that in BHI, which contains 27 mM glucose and also complex host glycans, which are rich in non-glucose sugars including galactose, galactose catabolism is acting as a repressor for the biofilm formation. Currently, we do not know the exact mechanism of the phenotype observed with the double deletion strain, but it may be due to the imbalance of metabolic precursors important for biofilm formation. It was reported that metabolic regulation plays a central role in the adaptation from the planktonic to biofilm phenotype[40]. Indeed, biofilm-forming pneumococci had reduced synthesis of enzymes of the glycolytic pathway and proteins involved in translation, transcription, and virulence, while proteins with a role in pyruvate, carbohydrate, and arginine metabolism increased significantly[40]. Alternatively, in the absence of galactose catabolic pathways in the double mutant, catabolite repression is released, and this may also be important for excess biofilm formation. Overall, our data demonstrates that the processing of carbohydrates plays a key role in the regulation of biofilm development, and further studies are needed to understand the underlying mechanisms for the contribution of galactose catabolic pathways on biofilm synthesis

The main hypothesis underlying our current and previous studies is that galactose utilization potentiates pneumococcal virulence. This hypothesis conflicts with the fact that the pneumococcus is a commensal of respiratory mucosa, but in this study, we found that sialic acid catabolism suppresses both catabolic pathways of galactose and the galactose transporters of the pneumococcus. Thus, by suppressing galactose utilization, sialic acid catabolism could maintain the

commensal existence of pneumococci in the respiratory tract. Therefore, by studying the conditions that remove sialic acid catabolism's repressor action on galactose catabolic pathways, we could identify the factors giving rise to shift from colonization to invasiveness. This hypothesis will be the subject of future research.

Our results support the hypothesis that the utilization of carbon sources, driven by the physical and chemical environment of a given niche, plays an important role in the pneumococcal switch from colonization to invasiveness by effecting its physiology and virulence. These data are in agreement with a recent study that showed that in galactose-rich culture conditions, mimicking the environment of the respiratory tract, the pneumococci had reduced metabolic activity and a lower growth rate, characterized by mixed acid fermentation with increased $H_2O_2$ production. On galactose, the bacterium was in a carbon-catabolite repression-derepressed state relative to *S. pneumoniae* grown on glucose glucose-rich medium. The glucose-rich medium resembles the blood nutritional environment[36], and the pneumococci form shorter chains, produced more capsules, are less adhesive, and are more resistant to macrophage killing in an opsonophagocytosis assay. Thus, there is building evidence that nutrient source is a major contributor to an array of virulence-associated phenotypes.

Finally, our work using Gal-3F showed that the pneumococcal catabolic pathways can be a viable anti-infective target, providing a potential lead for a new class of antimicrobials in the context of rising antibiotic resistance among clinical strains of the pneumococcus[41]. In the future, we plan to synthesize a diverse set of inhibitors acting on the galactose catabolic pathways and test them in vitro and in vivo to determine their potential to control pneumococcal infections.

## Methods

### Bacterial strains and growth

The wild-type *S. pneumoniae* type 2 D39 strain or genetic mutants was used in this study. The pneumococcal strains were grown at 37 °C either in Brain Heart Infusion (BHI) broth (Oxoid) or on Blood Agar Base (BAB) (Oxoid) supplemented with 5% (v/v) defibrinated horse blood. We also used chemically defined medium (CDM)[14] containing 55 mM glucose, or galactose *Escherichia coli* strain DH5α was used for plasmid propagation. It was grown in Luria broth (LB) in a shaking incubator at 37 °C or on Luria agar plates (LA). When necessary, the growth medium was supplemented with 100 μg/ml spectinomycin, 100 μg/ml ampicillin, 50–200 μg/ml kanamycin, and 15 μg/ml tetracycline, as appropriate. Bacterial stocks were stored at − 80 °C in 15% (v/v) glycerol until required.

**Construction of mutants and reporter strains.** The construction of ΔgalK, ΔlacD and ΔgalKΔlacD strains was done using mariner mutagenesis[14], and Δrgg144, Δrgg1518 were constructed using the SOEing method[13,24–26]. PgalK::luc and PlacA::luc were constructed using pPP3[34] and the primers listed in Supplementary Table 1. The LacZ reporter constructs, PgalK::lacZ, PlacA::lacZ, were made by amplifying the upstream regions containing the putative promoters of galK and lacA using the primers listed in Supplementary Table 1 and cloning into pPP2[42]. The constructs then were transformed into wild type or Δrgg144, Δrgg1518. PCR amplification and genetic transformations were done using standard methods[25,26,43].

### β-galactosidase assays

β-galactosidase activity was determined by using the Miller method[28,43]. Three milliliters of pneumococcal culture pellets grown to mid-exponential phase in CDM with the desired sugars were suspended in 3 ml chilled Z buffer (0.80 g Na2HPO4.7H2O, 0.28 g NaH2-PO4.H2O, 0.5 ml 1 M KCl, 0.05 ml 1 M MgSO4, 0.175 ml B-mercaptoethanol (BME), pH 7.0). The optical density of resuspended cells was recorded at OD600 nm, using Z-buffer as blank. Then, 500 μl of the cells were further diluted by 500 μl Z-buffer in a 1:1

ratio and permeabilized with one drop of Triton X-100. In the next step, the samples were incubated for 10 min at 30 °C, then 200 μl of ONPG (O-Nitro phenyl β-D-galactopyranoside) (4 mg/ml) solution was added to the samples, and the tubes were incubated at 30 °C. Subsequently, the reaction was stopped by the addition of 400 μl of 1 M Na2CO3, when a visible yellow color had developed, and the reaction time was recorded. Finally, the samples were centrifuged for 5 min at $14462 \times g$ a benchtop centrifuge and the A420 nm of the supernatants was measured. β- galactosidase activity unit was determined as nanomoles of p-nitrophenol released per unit of time per unit of volume of cell suspension.

### Capsule quantification

The effect of different galactose catabolic pathways on capsule production was quantified by the assay of glucuronic acid[26,44]. To do this, the pneumococcal strains, ΔgalK, ΔlacD and ΔgalKΔlacD, and the wild type, were obtained through lung lavage (see below) and added to 100 μl of 1% (v/v) Zwittergent 3-14 detergent (Sigma-Aldrich) in 100 mM citric acid (pH 2.0). The mixture was then incubated at 50 °C for 20 min and the polysaccharides were precipitated in 1 ml absolute ethanol. The pellet was dissolved in 200 μl of distilled water and mixed with 1.2 ml of 12.5 mM borax (Sigma-Aldrich) in $H_2SO_4$. The mixture was boiled at 100 °C for 5 min, cooled to room temperature, and mixed with 20 μl of 0.15% (w/v) 3-hydroxydiphenol (Sigma-Aldrich). Absorbance was measured at 520 nm and the glucuronic acid was quantified by comparison to a standard curve generated with known concentrations of glucuronic acid and normalized to $10^8$ CFU.

### Biofilm assay and crystal violet staining

Strains were grown in BHI until $OD_{600} = 0.05$ and then 2 mL culture were each seeded in 12-well plates in triplicate. Incubation was carried out at 37 °C, 5% $CO_2$ for 24 h. The supernatant was aspirated, and biofilms were stained with 0.1% crystal violet solution for 10 min at RT. Each well was then washed twice with PBS and let to dry. Pictures were taken and quantification was carried out using 70% Ethanol for resuspension (10 min at RT). Optical densities were measured. For the growth curve strains were inoculated in 20 mL BHI and grown at 37 °C, 5% $CO_2$. Once they reached $OD_{600} = 0.05$, measurements were taken every 30 min.

### Real time quantitative reverse transcriptase PCR (qRT-PCR)

RNA was extracted from the pneumococci obtained through lung lavage (see below) using the TRIZOL method[32]. SuperScript III reverse transcriptase (Invitrogen) was used to perform the first-strand cDNA synthesis, according to the manufacturer's instructions, using DNase-treated RNA (~ 500 ng) and random primers (Invitrogen), in 20 μl. qRT-PCR was done using a SensiMixTM SYBR® Hi-ROX kit (Bioline, UK). The transcription level of selected genes was normalized to the expression level of the housekeeping gyrB gene (Supplementary Table 1). The fold difference in expression was calculated by the CT method[45].

### RNAseq analysis

For gene expression, 10 ml *S. pneumoniae* culture grown to early-exponential phase in BHI was pelleted. The pellet was washed once with PBS, pH 7.0. Then, 2 ml culture was incubated in CDM supplemented either with 25 mM galactose or 3.5 mM ManNAc for 30 min separately, or 15 and 30 min with galactose and ManNAc together. The NucleoSpinRNA II kit (Macherey-Nagel, Germany) was used for RNA extraction. Frozen RNA samples were sent to the Institute of Applied Genomics (University of Udine, Italy) for RNA-seq analysis using an Illumina Genome Analyzer II platform (Illumina). RNA-Seq fastq files were trimmed using ERNE, read mapping to the reference genome D39 was performed using Tophat, and transcript abundance and differential expression analyses were analyzed using Cufflinks and the R package cummeRbund. Three independent replicas were used for

each sample. The RNA-seq data were deposited in the Gene Expression Omnibus database with accession code GSE246424.

## Infection experiments

Mouse studies were performed under the project (permit no. PP0757060) and personal (permit no. 80/10279) licenses according to the United Kingdom Home Office guidelines under the Animals Scientific Procedures Act 1986, and the University of Leicester ethics committee approval. The protocol used was approved by both the U.K. Home Office and the University of Leicester ethics committee. Where specified, the procedures were done under anesthetic with isoflurane. Animals were kept in individually ventilated cages in a controlled environment (temperature: $21 \pm 2$ °C and relative humidity (RH): 55%rh $\pm 10$) with 12 h dark and 12 h light cycles, and were regularly monitored after infection to reduce suffering.

CD1 and MF1 outbred mouse strains between the ages of 8 to 10 weeks were used. The mice were bred in-house at the University of Leicester. To prepare a standard inoculum, pneumococci were grown to OD600 1-4 to 1.6 in BHI supplemented with 20% (v/v) calf serum. For inoculum propagated in CDM supplemented either with glucose or galactose, cultures were grown to OD600 0.3-0.4, corresponding to the early exponential phase. Cultures were maintained stationary at 37 °C for incubation. Samples were kept at $-80$ °C until needed. For the pneumonia model, mice were anesthetized using 2.5% (v/v) isoflurane (Isocare) over oxygen (1.4–1.6 liters/min) and the infective dose was given intranasally in 50 µl PBS drop-wise[13,25]. Mice were scored for the signs of disease (starry coat, hunched, and lethargic) for 7 days[13,23,25]. When they reached a lethargic state, they were culled. Time to reach a lethargic state was considered as the survival time.

For imaging, mice were infected with approximately $5 \times 10^6$ CFU/mouse in 50 µl PBS, under light anesthesia using 2.5% (v/v) isoflurane over oxygen (1.4–1.6 liters/min). At predetermined time points, preselected mice were anesthetized and luciferin (150 mg/kg) administered subcutaneously. The animals were imaged using the IVIS® Spectrum in vivo imaging system (Perkin Elmer) over 20 min at one-minute intervals. The peak signal was determined and data were analyzed by an unpaired $t$ test. Immediately after imaging, mice were killed to dissect their lungs and the pneumococcal counts were determined in the lung homogenates to normalize the signal levels against the colony-forming units[34].

Pneumococci also were collected from mouse lungs after intranasal infection via lavage to measure pneumococcal gene expression and assayed for glucuronic acid level to quantify capsule synthesis. To do this, mice were infected intranasally with pneumococcal strains as described above. Four hours after infection, animals were killed with isopentane and the lung lavage was obtained using 1 ml PBS. For gene expression studies, the lavage samples were centrifuged at $12470 \times g$ in a benchtop centrifuge, the supernate was removed and the pellet was resuspended in 0.5 ml Trizol reagent (Thermo Fisher Scientific) for RNA extraction. For glucuronic acid assay, the lavage samples were kept at $-80$ °C until needed.

## Data analysis

Data analysis was done using GraphPad Prism version 9.4 (Dotmatics).

## Reporting summary

Further information on research design is available in the Nature Portfolio Reporting Summary linked to this article.

## Data availability

All data supporting the paper's conclusions are included in the main text and supplementary material file. The RNA-seq data is available from the Gene Expression Omnibus (GEO) repository with the primary accession code GSE246424. Source data are provided with this paper.

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

## Acknowledgements
We kindly acknowledge the technical support provided by the DBS in Leicester, UK, for the in vivo work, and Ms Lucy Onions for IVIS studies. We are grateful for the support from the NIH (R01 AI139077-01A1 to NLH and R01 AI135060-01A1 to AU).

## Author contributions
B.O.K., O.G., K.M.B., and M.H. performed the experiments and analyzed the results, D.G.G. and A.T.U. provided experimental tools, N.L.H., M.R.O., P.W.A., and H.Y. conceived the project. H.Y. wrote the paper with input from all other authors.

## Competing interests
The authors declare no competing interests.
