## [Peer Review File · Nature Communications]

Environmental and genetic regulation of *Streptococcus pneumoniae* galactose catabolic pathwaysReviewer #1 (Remarks to the Author):

Kareem and colleagues explore the role of the two major galactose catabolic pathways in the opportunistic airway pathogen *Streptococcus pneumoniae*. They demonstrate that different environmental signals regulate the two pathways, with the tagatose pathway triggered at low galactose concentrations and at febrile temperatures, whilst the Leloir pathway requires higher galactose concentrations and is unaffected by temperature change. The tagatose pathway is rapidly induced during pneumococcal pneumonia, whilst the Leloir pathway was not recorded until much later. Products of sialic acid metabolism inhibit both pathways and the authors provide proof of principle data demonstrating that galactose catabolism is a druggable pathway.

The paper is well written and the introduction and discussion sections provide a balanced view of the literature. The authors' central hypotheses - that pneumococcal metabolism is closely linked to virulence and that metabolic switches might underpin the commensal-pathogen conversion - is a compelling one. A detailed mechanistic understanding of the regulation of pneumococcal metabolism under environmentally-relevant conditions is sorely needed and I applaud the authors' efforts to address this knowledge gap.

I found most of the presented data convincing, and the luciferase-reporter strains have been utilised smartly throughout. I have some questions regarding experimental details and data interpretation, which I hope the authors will find useful:

1. The difference in the outcomes of the mice intra-nasally infected with pneumococci prepared in glu or gal are striking, but how confident are the authors that this outcome relates solely to a priming of the bacteria for the galactose-rich airway environment? I think this interpretation needs to be considered in more depth. For example, the authors highlight that galactose pathways contribute to capsule production. Did they compare the capsules of pneumococci grown in glu vs gal? Perhaps the gal-grown bacteria are more virulent due to a thicker capsule at the time of infection. Capsule analysis could also be performed on the deletion strains used in Fig 6, prior to their use in infection models, to confirm that they are comparable to the parental strain.
2. At line 186, the authors highlight that D39 is unable to catabolize sialic acid due to mutations in the neuraminidase gene. How do they square this with their previous finding of nanA-dependent growth on mucin (PMID: 18053067)? Is the removal of terminal sialic residues required merely to access linked galactose? Here, the authors go on to use ManNAc to demonstrate a substrate-induced metabolic switch away from galactose catabolic pathways. Have they tried using sialic acid and comparing D39 to a strain with a functional neuraminidase? It would be interesting to know whether the initial dissimilation step is required for the metabolic switch. This relates to discussion at line 395 where the authors suggest sialic acid catabolism is the main brake on the shift from colonisation to invasiveness. Would the prediction not then be that D39 should act as a primary pathogen?
3. I like the idea of galactose catabolism acting as a virulence switch, but I wonder if the various permutations of this could be discussed more thoroughly. For example, the authors argue that galactose utilisation needs to be suppressed to prevent virulence (line 394) but also that galactose acts as a main carbohydrate source during carriage (line 331). Yet the commensal-pathogen switch within nasopharynx remains a rare event. Do the authors ascribe this to the inhibitory effect of glycan sialic acids, and if so, then where and when does galactose utilisation become critical? I don't expect the authors to have all the answers for this, but would be interested to get their thoughts on why it might be advantageous for pneumococci to tie virulence so closely to the utilisation of a carbohydrate source that is associated with their commensal niche.
4. The phenotype of the double deletion strain in Figure 4 is striking. How do the authors interpret this, given the likely suppression of both lac and gal under the high-glucose conditions of BHI?
5. How do the authors account for the high level of induction of the tagatose pathway in the pneumonia model, given that their previous data suggested suppression under anaerobic conditions? This is touched on in the discussion (line 382), but I didn't follow the logic. What is meant by 'basal section of respiratory tract'? Did the authors conduct similar experiments in a

colonisation/carriage model, where glucose is scarce, but oxygen is available? From the image in Fig 5, the luciferase reporter assay appears sensitive enough to detect expression in nasopharynx. The last sentence of the results section highlights that galactose is the main carbohydrate source in carriage, so this additional experiment might provide a nice adjunct to their presented work.

6. Legends are lacking detail on the statistical test performed (e.g. Fig 2, Fig 7) and on multiple comparison correction (e.g. Fig 4) . Asterisks denote different levels of significance in different places (e.g. ** $p < 0.005$ in Fig 4, ** $p < 0.01$ in Fig 7).

Minor suggestions/queries

- In the second half of the abstract, where the in vivo results are discussed, the environmental context (i.e. niche) is not described. The timing of induction of the galactose catabolic pathways would presumably differ in different infection contexts, so it would be useful to know here whether carriage or a disease state is being considered.

- In Fig 1, what was the rationale for the choice of 55mM as the concentration of glucose and galactose for use in these experiments? In the discussion, the authors quite rightly focus on the sugar profile of airway environments, but the concentrations used in this experiment would appear to be way above what might be expected in the airways, where glucose has been reported to be at concentrations of less than 1mM (PMID: 22878875) and shown to be further depleted during pneumococcal pneumonia (PMID: 37669280).

- Line 241: 'Decreased' should read 'decrease'.

- Line 234, line 272 and line 286: There is some inconsistency in how the strains are referred to in the text. i.e. $\Delta\text{lacA}\Delta\text{galK}$ vs $\Delta\Delta\text{galKlacA}$.

- Line 297: I'm not clear what is meant by "possibly due to the occurrence of transcription and translation times". Could this sentence be reworded?

- In Fig 3, the temperature-dependent regulation of the tagatose pathway is an interesting finding. Did any of the genes of the pathway come up in the authors' previous analysis of temperature-dependent regulation of pneumococcal metabolism (PMID: 34491792)?

- In Fig 7, I can see that the data are normalised for CFU in panel D, but it would be useful for the reader to understand whether the Δrgg144 and $\Delta\text{rgg1518}$ strains are attenuated in virulence (i.e. is there a CFU difference between strains).

Reviewer #2 (Remarks to the Author):

Kareem and colleagues present findings that outline the environmental and genetic regulation of *Streptococcus pneumoniae* galactose catabolism. This is an important topic as it helps to explain how this lactic acid producing bacteria is able to transition from the airway, as both a commensal and pathogen, to invasive disease sites. In general, the authors are to be lauded on their work and the clarity of the document, which neatly explains a very complex and interwoven metabolic network.

Introduction, the authors do an excellent job of providing rationale for the project and explaining the impact of galactose vs sialic acid vs glucose and the known roles for the Leloir and tagatose pathways.

Introduction, P5, the authors cite a publication that a type 2 strain (D39) produces greater capsule when grown on galactose versus glucose [Carvalho et al]. This sentence is worded in somewhat conclusive fashion – suggesting it holds up across all *S. pneumoniae*. Yet work by Troxler [PMID: 31594867] suggests capsule production in response to glucose is serotype dependent. What is more, a study by Im [PMID 34748366] shows glucose grown type 4 pneumococci have more

capsule than a galactose grown equivalent. The sentence should be reworded so as not to be as conclusive. In the discussion, the complexity and inconsistency in the literature is an important consideration that should be communicated to the reader.

Results, P6, given the prior statement, and since the authors do it for other conditions (Fig 6), it would be good to measure capsule amount in D39 grown in galactose and glucose conditions tested in Fig 1.

Results, P8, since D39 lacks neuraminidase lyase, investigators use ManNAc as surrogate for sialic acid and show that this is a preferred carbon source over galactose. RNAseq data supports this. RT-PCR of the same genes in another sufficient strain of *S. pneumoniae* administered ManNAc as well as sialic acid, would strengthen this conclusion.

Results, P10, biofilm experiments should include galactose and glucose conditions tested in Fig 1.

Discussion, p14. Previously mentioned manuscript by Im et al. suggests that growth in galactose or glucose impacts relative fitness of the bacterium. How this finding fits into the authors conclusions merits discussion.

This is a very strong paper and the results are convincing. However, and as supported by other literature, there are circumstances where the observations proposed may not be applicable to other strains. Mutations in key metabolic genes (such as D39 has in neuraminidase lyase), metabolic demands of other serotypes are likely to influence the results seen in other strains and serotypes that nonetheless are able to cause disease. Authors should be indicate that readers must take into consideration strain specific features when interpreting data.

Reviewer #3 (Remarks to the Author):

The study by Kareem et al., presents an investigation of the regulation of galactose utilisation in *S. pneumoniae*. The authors have shown that the tagatose and Leloir pathways are differentially responsive to galactose abundance in addition to other environmental conditions. This has been aided by effective use of in vivo reporters such as the luc system. However, while the study has made some interesting observations regarding the activation and regulation of the galactose catabolism pathways, the overall lack of mechanistic insight to explain these observations is problematic. This is particularly evident for the multiple instances of conflicting data within the manuscript, which are justified by vague statements about complexity of regulation.

Specific concerns are below:

The choice to use D39 for this study is a little puzzling. The authors have noted that D39 is not able to utilise sialic acid, and this is supported by the ManNAc RNA seq data, which suggests that catabolism of sialic acid will be prioritised over galactose. This would indicate that the data generated have limited relevance for aspects of native infection including biofilm formation. There is of course a benefit to using well-defined laboratory strains, but given that the authors were specifically investigating temporal activation of galactose pathways, this would appear to be the incorrect experimental system for this work.

The interplay between galactose catabolism and biofilm formation is well established in the literature but is largely not supported by the data generated in this study. Despite the authors stating that capsule biosynthesis relies on precursors from galactose catabolism, the $\Delta galK\Delta lacA$ strain produced more biofilm in vitro than the WT. These experiments were also conducted in BHI, which almost guarantees that glucose was being used as the carbon source, subverting the need for galactose utilisation. The in vitro data was also in direct contrast to the in vivo glucuronic acid assay. A Δcps strain should be included both in vitro and in vivo to confirm that these experimental systems are accurately reporting on capsule abundance.

The in vivo qRT-PCR data was another source of inconsistency and appeared to be under explored. For example, what were the recovered CFUs for the mutant strains compared to the WT? Are there differences in virulence/host clearance that manifest within the first 4 hours of infection? This could be particularly interesting for the Δ lacD strain, which is relying on the Leloir pathway at much earlier timepoints.

There were also some results that did not appear to match the narrative, such as the down-regulation of pflA and pflB. If the bacterium was indeed using galactose as the primary carbon source, then the genes required for mixed acid fermentation should have been up-regulated in the single mutants (given the suggested redundancy in the Leloir and tagatose pathways). This may suggest that homolactic fermentation was still being primarily utilised and therefore confounding the findings and conclusions regarding galactose utilisation.

The rationale for investigating the rgg144 and rgg1518 systems is also not well defined and very little discussion of how these systems intersect with galactose catabolism is offered past reference to earlier works. The impact of removing rgg144 on lacA expression in vitro is quite striking, however this is not replicated in vivo. While this is not unusual for metabolic pathways due to innate complexities in regulation, this finding is not further investigated, or even discussed, and thus has no bearing on the conclusions of the study overall. This appears to be a missed opportunity given the expertise of the authors in this research area.

REVIEWER COMMENTS

Reviewer #1 (Remarks to the Author):

Kareem and colleagues explore the role of the two major galactose catabolic pathways in the opportunistic airway pathogen *Streptococcus pneumoniae*. They demonstrate that different environmental signals regulate the two pathways, with the tagatose pathway triggered at low galactose concentrations and at febrile temperatures, whilst the Leloir pathway requires higher galactose concentrations and is unaffected by temperature change. The tagatose pathway is rapidly induced during pneumococcal pneumonia, whilst the Leloir pathway was not recorded until much later. Products of sialic acid metabolism inhibit both pathways and the authors provide proof of principle data demonstrating that galactose catabolism is a druggable pathway.

The paper is well written and the introduction and discussion sections provide a balanced view of the literature. The authors central hypotheses - that pneumococcal metabolism is closely linked to virulence and that metabolic switches might underpin the commensal-pathogen conversion - is a compelling one. A detailed mechanistic understanding of the regulation of pneumococcal metabolism under environmentally-relevant conditions is sorely needed and I applaud the authors' efforts to address this knowledge gap.

I found most of the presented data convincing, and the luciferase-reporter strains have been utilised smartly throughout. I have some questions regarding experimental details and data interpretation, which I hope the authors will find useful:

Authors' response to Reviewer 1: We are very pleased with the reviewer's appreciation of our study and for the constructive suggestions. We have addressed all the points raised by the reviewer.

Q1. The difference in the outcomes of the mice intra-nasally infected with pneumococci prepared in glu or gal are striking, but how confident are the authors that this outcome relates solely to a priming of the bacteria for the galactose-rich airway environment? I think this interpretation (a.) needs to be considered in more depth. For example, the authors highlight that galactose pathways contribute to capsule production. Did they compare the capsules of pneumococci grown in glu vs gal? (b.) Perhaps the gal-grown bacteria are more virulent due to a thicker capsule at the time of infection. Capsule analysis could also be performed on the deletion strains used in Fig 6, prior to their use in infection models, to confirm that that they are comparable to the parental strain.

A1.) As requested by the reviewer we determined the glucuronic acid level as an assay for capsule synthesis, of the wild type D39 strain on glucose and galactose. Indeed, on galactose, the pneumococcus produced more glucuronic acid than on glucose. This point has been included in the modified manuscript as follows (ln. 153-161).

'The pneumococcal cultures grown on galactose are known to produce more capsule than those grown on glucose¹⁵. Therefore, to determine whether the potency of virulence associated with the galactose-grown cultures is linked to a higher level of capsule synthesis, we evaluated the capsule level of the pneumococci on glucose versus galactose growth. The results showed that on galactose ($10.5 \pm 0.2 \mu\text{M}/10^8 \text{ CFU}$), type 2 D39 strain synthesized more capsule than on glucose ($3.8 \pm 0.3 \mu\text{M}/10^8 \text{ CFU}$). This shows that, in addition to the metabolic adaptation that allows efficient utilization of

galactose, a higher capsule level could also be responsible for potentiating the impact of galactose growth.

We also determined the capsule levels for $\Delta galK$ ($63 \pm 0.5 \mu\text{M}/10^8$ CFU, $n=4$), $\Delta lacD$ ($58 \pm 0.5 \mu\text{M}/10^8$ CFU, $n=4$) and $\Delta galK/lacD$ ($57 \pm 0.2 \mu\text{M}/10^8$ CFU, $n=4$) mutants on BHI prior to their use in infection model (In 323-328). The results showed no difference in capsule (glucuronic acid) levels compared to the wild type ($64 \pm 0.3 \mu\text{M}/10^8$ CFU, $n=4$). As an assay control, we also included Δcps ($39 \pm 0.1 \mu\text{M}/10^8$ CFU, $n=4$), which had significantly less capsule synthesis compared to the wild type ($p < 0.05$). Therefore, the decreased capsule levels in the mutants $\Delta galK$, $\Delta lacD$ and $\Delta galK/lacD$ recovered from the respiratory tract was very likely due to inefficient utilisation of galactose. These data have been included in the modified manuscripts as follows (In 330-335):

'Before infection, no difference in capsule levels was seen among the pneumococcal strains propagated on BHI: wild type ($64 \pm 0.3 \mu\text{M}/10^8$ CFU, $n=4$), $\Delta galK$ ($63 \pm 0.5 \mu\text{M}/10^8$ CFU, $n=4$), $\Delta lacD$ ($58 \pm 0.5 \mu\text{M}/10^8$ CFU, $n=4$) and $\Delta galK\Delta lacD$ ($57 \pm 0.2 \mu\text{M}/10^8$ CFU, $n=4$) ($p > 0.05$). As an assay control, we also included Δcps ($39 \pm 0.1 \mu\text{M}/10^8$ CFU, $n=4$), which had significantly less capsule synthesis compared to the wild type ($p < 0.05$).

2. (A.) At line 186, the authors highlight that D39 is unable to catabolize sialic acid due to mutations in the neuraminatase lyase gene. How do they square this with their previous finding of nanA-dependent growth on mucin (PMID: 18053067)? Is the removal of terminal sialic residues required merely to access linked galactose? Here, the authors go on to use ManNAc to demonstrate a substrate-induced metabolic switch away from galactose catabolic pathways. Have they tried using sialic acid and comparing D39 to a strain with a functional neuraminatase lyase? It would be interesting to know whether the initial dissimilation step is required for the metabolic switch. This relates to discussion at line 395 where the authors suggest sialic acid catabolism is the main brake on the shift from colonisation to invasiveness. Would the prediction not then be that D39 should act as a primary pathogen?

A2. (A) In our previous publication (PMID: 18053067), we indeed found that in the absence of NanA, the pneumococcus was unable to grow on pork gastric mucin (PGM). We showed that pre-treatment of fetuin, complex host glycoprotein composed of O- and N-linked glycoproteins, with NanA significantly increases galactose release. These findings clearly indicate the significant role of NanA on initiation of galactose release. In the absence of sialidase activity, galactosidases cannot 'see' their substrate, hence the pneumococcus releases less galactose and the growth is attenuated. However, NanA's role is not limited with galactose release alone. Depending on the assay systems (in vitro, ex vivo or in vivo) several other important roles have been attributed to the pneumococcal NanA such as stimulation of mucus production in vivo (PMID: 33819312), host Inflammation and cell death (PMID: 33777832), complement triggering and hemolytic activity (PMID: 31297339), and biofilm formation (PMID: 28096183).

The impact of sialic acid on TIGR4 strain (serotype 4), which has a functional neuraminatase lyase, was tested by growth and gene expression studies (In. 213-223). The result showed that similar to ManNAc, sialic acid induced metabolic switch away from galactose catabolic pathways (SFigure 1), indicating that the initial dissimilation step is not required for the metabolic switch. In addition, the addition of either 3.5 mM ManNAc or sialic acid to the pneumococcal cultures grown on galactose led to a significant decrease in the expression of *galK* (3.4 ± 0.07 and 6.6 ± 0.03 , $n=4$ for ManNAc and sialic acid, respectively) and *lacD* (4.5 ± 0.15 and 2.8 ± 0.25 , $n=4$ for ManNAc and sialic acid, respectively) compared to the growth on galactose. Therefore, our observation is reproducible and

not strain-specific. These points have been included in the modified manuscripts as follows (ln.213-223):

‘To determine whether we could reproduce our results with a strain that has an intact neuraminidase gene, we used serotype 4 TIGR4 strain. When we added 3.5 mM sialic acid on galactose grown pneumococci, a substrate-induced metabolic switch could also be demonstrated with the TIGR4 strain by growth studies (SFigure 1). Moreover, the addition of either 3.5 mM ManNAc or sialic acid to the pneumococcal cultures grown on galactose led to a significant decrease in the expression of *galK* (3.4 ± 0.07 and 6.6 ± 0.03 , $n=4$ for ManNAc and sialic acid, respectively) and *lacD* (4.5 ± 0.15 and 2.8 ± 0.25 , $n=4$ for ManNAc and sialic acid, respectively) compared to the growth on galactose. Therefore, our observation that sialic acid addition decreases the expression of both galactose catabolic pathways and leads to a substrate-induced metabolic switch is reproducible and not strain-specific.’

3. I like the idea of galactose catabolism acting as a virulence switch, but I wonder if the various permutations of this could be discussed more thoroughly. For example, the authors argue that galactose utilisation needs to be suppressed to prevent virulence (line 394) but also that galactose acts as a main carbohydrate source during carriage (line 331). Yet the commensal-pathogen switch within nasopharynx remains a rare event. Do the authors ascribe this to the inhibitory effect of glycan sialic acids, and if so, then where and when does galactose utilisation become critical? I don't expect the authors to have all the answers for this, but would be interested to get their thoughts on why it might be advantageous for pneumococci to tie virulence so closely to the utilisation of a carbohydrate source that is associated with their commensal niche.

A3. We are pleased that the reviewer liked our idea about the sialic acid's role in commensal to the pathogen switch. While we did not study when this switch may be happening, based on the available literature and our data, we can put forward environmental and host related factors or combination of both, as the likely cause for the shift. We showed in this study that the induction of galactose catabolic pathways is influenced by environmental factors, including galactose and oxygen concentration, and temperature, and the activation of the individual pathways require different environmental conditions. We hypothesise that the phenotype conferred by each galactose catabolic pathway would be unique at different stages of infection depending on the host's immune status. This affects the microbial factors such as the synthesis of virulence determinants and pneumococcal adaptation, including their response to sialic acid, and host response to the pneumococcus.

Q4. The phenotype of the double deletion strain in Figure 4 is striking. How do the authors interpret this, given the likely suppression of both *lac* and *gal* under the high-glucose conditions of BHI?

A4. Our data show that in glucose rich BHI, galactose catabolism is acting as a repressor for the biofilm formation. Currently we do not know the exact mechanism of the phenotype observed with the double deletion strain but it may be due to the imbalance of metabolic precursors important for biofilm formation. Overall, our data demonstrates that the processing of carbohydrates plays a key role in the regulation of biofilm development.

Q5. (A) How do the authors account for the high level of induction of the tagatose pathway in the pneumonia model, given that their previous data suggested suppression under anaerobic conditions? This is touched on in the discussion (line 382), but I didn't follow the logic. What is meant by 'basal section of respiratory tract'? Did the authors conduct similar experiments in a

colonisation/carriage model, where glucose is scarce, but oxygen is available? From the image in Fig 5, the luciferase reporter assay appears sensitive enough to detect expression in nasopharynx. The last sentence of the results section highlights that galactose is the main carbohydrate source in carriage, so this additional experiment might provide a nice adjunct to their presented work.

A5. We want to make it clear that the respiratory tract is not an anaerobic environment. It has been reported that the level of oxygen is around 13.6% (Rafael S. Carel, Health Aspects of Air Pollution, from 'Handbook of Air Pollution From Internal Combustion Engines', Academic Press, 1998, pg. 42-64) in the lower respiratory tract and galactose containing host glycans are also synthesised by the goblet cells in the respiratory tract (Front Endocrinol (Lausanne) 2013 Vol. 4 Pages 129). Therefore, our pneumonia model is relevant for testing the induction level of individual galactose catabolic pathways because of the presence of galactose and oxygen. What is meant by the basal section of respiratory tract is the basal membrane. We corrected this (ln.458).

Q6. Legends are lacking detail on the statistical test performed (e.g. Fig 2, Fig 7) and on multiple comparison correction (e.g. Fig 4) . Asterisks denote different levels of significance in different places (e.g. **p<0.005 in Fig 4, **p<0.01 in Fig 7).

A6. The legends were added detail on the statistical tests performed. The same level of significance was represented with the consistent asterisk denotation.

Minor suggestions/queries

Q7.- In the second half of the abstract, where the in vivo results are discussed, the environmental context (i.e. niche) is not described. The timing of induction of the galactose catabolic pathways would presumably differ in different infection contexts, so it would be useful to know here whether carriage or a disease state is being considered.

A7. As suggested, we indicated the environmental context of the induction as 'the lungs after intranasal infection'(ln.45-46).

Q8. In Fig 1, what was the rationale for the choice of 55mM as the concentration of glucose and galactose for use in these experiments? In the discussion, the authors quite rightly focus on the sugar profile of airway environments, but the concentrations used in this experiment would appear to be way above what might be expected in the airways, where glucose has been reported to be at concentrations of less than 1mM (PMID: 22878875) and shown to be further depleted during pneumococcal pneumonia (PMID: 37669280).

A8. Historically, in our studies we have been using non-limiting (55mM) sugar concentration in chemically defined medium. We are aware that this concentration is well above galactose and glucose concentration found in the respiratory tract. The idea here was to obtain an infective dose with two metabolically distinct phenotypes to test the virulence potentiating impact of different sugars at similar biomass levels. We used non-limiting sugar concentration because at this concentration the biomass obtained with galactose and glucose growth is similar (Paixão et al. PLoS One. 2015 Mar 31;10(3):e0121042). When we used limiting (13 mM) and non-limiting concentrations (34 mM) of mannose, glucose, N-acetyl glucosamine or galactose, we found that the specific growth rate is independent of the initial substrate concentration, except for galactose which supports higher growth rates when the substrate is in excess. When the higher substrate concentration was used, galactose supported the highest final biomass, which was similar to that on glucose.

Q9. Line 241: 'Decreased' should read 'decrease'.

A9. This has been corrected as 'decrease' (ln. 266)

Q10. Line 234, line 272 and line 286: There is some inconsistency in how the strains are referred to in the text. i.e. $\Delta lacA\Delta galK$ vs $\Delta\Delta galK lacA$.

A10. Consistent abbreviation has been used to represent the double mutant throughout the text (ln. 298, 302, 321, 325, 332, 398).

Q11. Line 297: I'm not clear what is meant by "possibly due to the occurrence of transcription and translation times". Could this sentence be reworded?

A11. This sentence has been reworded as (ln337-339): 'The difference in gene expression of *cps2A* and capsule levels in $\Delta galK$, $\Delta lacD$ and $\Delta galK\Delta lacD$ may have occurred due to the differences in the length of time required for both transcription as well as posttranslational control'.

Q12. In Fig 3, the temperature-dependent regulation of the tagatose pathway is an interesting finding. Did any of the genes of the pathway come up in the authors' previous analysis of temperature-dependent regulation of pneumococcal metabolism (PMID: 34491792)?

A12. We did not see differential regulation of tagatose pathways genes in our previous work (PMID: 34491792). The likely reason for this is that we used glucose rather than galactose when we did our microarray analysis to determine differentially regulated genes at different temperatures.

Q13. In Fig 7, I can see that the data are normalised for CFU in panel D, but it would be useful for the reader to understand whether the $\Delta rgg144$ and $\Delta rgg1518$ strains are attenuated in virulence (i.e. is there a CFU difference between strains).

A13. We did not see any significant difference in the colony counts 4 hours post infection when we scanned the mice. We modified the sentence in the Figure 7 legend as follows: 'Animals were scanned at 4 h post-infection, when there was no difference in the recovered colony counts among the strains, *PlacA-luc:wt*: 4.45×10^6 , *PlacA-luc: $\Delta rgg144$* : 4.75×10^6 and *PlacA-luc: $\Delta rgg1518$* : 4.3×10^6 '.

Reviewer #2 (Remarks to the Author):

Kareem and colleagues present findings that outline the environmental and genetic regulation of *Streptococcus pneumoniae* galactose catabolism. This is an important topic as it helps to explain how this lactic acid producing bacteria is able to transition from the airway, as both a commensal and pathogen, to invasive disease sites. In general, the authors are to be lauded on their work and the clarity of the document, which neatly explains a very complex and interwoven metabolic network.

Introduction, the authors do an excellent job of providing rationale for the project and explaining the impact of galactose vs sialic acid vs glucose and the known roles for the Leloir and tagatose pathways.

Authors' response to Reviewer 1: We are very pleased with the reviewer's appreciation of our study and for the constructive suggestions. We have addressed all the points raised by the reviewer.

Q1. Introduction, P5, the authors cite a publication that a type 2 strain (D39) produces greater capsule when grown on galactose versus glucose [Carvalho et al]. This sentence is worded in somewhat conclusive fashion – suggesting it holds up across all *S. pneumoniae*. Yet work by Troxler [PMID: 31594867] suggests capsule production in response to glucose is serotype

dependent. What is more, a study by Im [PMID 34748366] shows glucose grown type 4 pneumococci have more capsule than a galactose grown equivalent. The sentence should be reworded so as not to be as conclusive. In the discussion, the complexity and inconsistency in the literature is an important consideration that should be communicated to the reader.

A1. We thank the reviewer for raising this point. We modified the text to indicate that our statement encompasses only type 2 D39 strain. The modifications included:

‘Importantly, the synthesis of the pneumococcal polysaccharide capsule, which is essential for virulence, is affected by galactose utilization in type 2 D39 strain.’ (ln. 110-112).

‘Critically, although not surprisingly, it has been shown that sugar catabolism is a determinant of the extent of pneumococcal encapsulation in a type 2 D39 strain, which produced a larger quantity of capsule when grown on galactose than on glucose’ (ln.114-117).

In addition, we discussed this point in detail as follows (ln. 405-415):

‘Capsule biosynthesis is a highly complex process in *S. pneumoniae* and can be influenced by the capsule type, promoter sequence (regulation) and carbon source^{34,35,36}. Troxler et al. (2019) showed that serotype 7F pneumococci synthesized significantly less capsule than other tested serotypes (6B, 6C, 9V, 15, and 23F) when cultured on glucose or sucrose³⁵. A recent study showed that TIGR4 strain produces more capsule when cultivated on 1 g/L glucose and 0.6 g/L sialic acid than on 0.6 g/L galactose and 0.3 g/L sialic acid³⁶. While differences in capsule level in TIGR4 on glucose and galactose can be due to the role of these sugars on capsule biosynthesis in this genetic background, it can be also due to sialic acid’s suppressive effect on galactose metabolism as we demonstrated in this study, or the differences in biomass between the two culture conditions¹⁴’.

Q2. Results, P6, given the prior statement, and since the authors do it for other conditions (Fig 6), it would be good to measure capsule amount in D39 grown in galactose and glucose conditions tested in Fig 1.

A2. As suggested by the reviewer, we have quantified the capsule level and this point has been incorporated in relation to the data in Figure 1 as follows (ln. 153-161):

‘The pneumococcal cultures grown on galactose are known to produce more capsule than those grown on glucose. Therefore, to determine whether the potency of virulence associated with the galactose-grown cultures is linked to a higher level of capsule synthesis, we evaluated the capsule level of the pneumococci on glucose and galactose growth. The results showed that on galactose ($1.05 \pm 0.02 \mu\text{M}/10^8 \text{ CFU}$), pneumococci synthesised more capsule than on glucose ($0.38 \pm 0.02 \pm 0.02 \mu\text{M}/10^8 \text{ CFU}$). This shows that, in addition to the metabolic adaptation that allows efficient utilisation of galactose, a higher capsule level could also be responsible for potentiating the impact of galactose growth’.

Q3. Results, P8, since D39 lacks neuraminidase, investigators use ManNAc as surrogate for sialic acid and show that this is a preferred carbon source over galactose. RNAseq data supports

this. RT-PCR of the same genes in another sufficient strain of *S. pneumoniae* administered ManNAc as well as sialic acid, would strengthen this conclusion.

A3. To address the reviewer's point, we have done both growth and RT-PCR experiments. Using a strain with an intact neuraminidase gene (serotype 4 TIGR4 strain), we showed that similar to ManNAc, sialic acid induced metabolic switch away from galactose catabolic pathways (SFigure 1). The relevant text has been added to the modified manuscript (ln.213-223):

'To determine whether we could reproduce our results with a strain that has an intact neuraminidase gene, we used serotype 4 TIGR4 strain. When we added 3.5 mM sialic acid on galactose grown pneumococci, a substrate-induced metabolic switch could also be demonstrated with the TIGR4 strain by growth studies (SFigure 1). Moreover, the addition of either 3.5 mM ManNAc or sialic acid to the pneumococcal cultures grown on galactose led to a significant decrease in the expression of *galK* (3.4 ± 0.07 and 6.6 ± 0.03 , $n=4$ for ManNAc and sialic acid, respectively) and *lacD* (4.5 ± 0.15 and 2.8 ± 0.25 , $n=4$ for ManNAc and sialic acid, respectively) compared to the growth on galactose. Therefore, our observation that sialic acid addition decreases the expression of both galactose catabolic pathways and leads to a substrate-induced metabolic switch is reproducible and not strain-specific'.

Q4. Results, P10, biofilm experiments should include galactose and glucose conditions tested in Fig 1

A4. It would have been ideal to assess the biofilm levels on glucose and galactose. However, $\Delta galK$, $\Delta lacD$ and $\Delta galK\Delta lacD$ mutants cannot grow on galactose in chemically defined medium as we reported before (PMID: 25826206)

Q5. Discussion, p14. Previously mentioned manuscript by Im et al. suggests that growth in galactose or glucose impacts relative fitness of the bacterium. How this finding fits into the authors conclusions merits discussion.

A5. As suggested, we discussed Im et al. findings as follows (ln. 473-485):

'Our results support the hypothesis that utilization of carbon source, driven by the physical and chemical environment of a given niche, plays an important role in the pneumococcal switch from colonization to invasiveness by effecting its physiology and virulence. These data are in agreement with a recent study that showed that in galactose rich culture condition, mimicking the environment of respiratory tract, the pneumococci had reduced metabolic activity and a lower growth rate, characterized by mixed acid fermentation with increased H_2O_2 production. On galactose the bacterium was in a carbon-catabolite repression-derepressed state relative to *S. pneumoniae* grown on glucose rich medium. The glucose rich medium, resembles the blood nutritional environment³⁶, and the pneumococci form shorter chains, produced more capsule, are less adhesive, and are more resistant to macrophage killing in an opsonophagocytosis assay. Thus, there is building evidence that nutrient source is a major contributor to an array of virulence-associated phenotypes.'

Q6. This is a very strong paper and the results are convincing. However, and as supported by other literature, there are circumstances where the observations proposed may not be applicable to other strains. Mutations in key metabolic genes (such as D39 has in neuraminidase), metabolic demands of other serotypes are likely to influence the results seen in other strains and serotypes that nonetheless are able to cause disease. Authors should be indicate that readers must take into consideration strain specific features when interpreting data.

A6. We thank the reviewer for the positive comments. We agree with the author about the strain specific effect of our observations. Therefore, in the revised manuscript, we addressed this point experimentally and also covered in the discussion. Experimentally, we showed that our observations are reproducible with another pneumococcal strain, TIGR4 (In. 213-223). In addition, we discussed the available literature to indicate that the degree of encapsulation may be strain and carbohydrate specific (In.405-415).

Reviewer #3 (Remarks to the Author):

The study by Kareem et al., presents an investigation of the regulation of galactose utilisation in *S. pneumoniae*. The authors have shown that the tagatose and Leloir pathways are differentially responsive to galactose abundance in addition to other environmental conditions. This has been aided by effective use of in vivo reporters such as the luc system. However, while the study has made some interesting observations regarding the activation and regulation of the galactose catabolism pathways, the overall lack of mechanistic insight to explain these observations is problematic. This is particularly evident for the multiple instances of conflicting data within the manuscript, which are justified by vague statements about complexity of regulation.

Specific concerns are below:

Q1. The choice to use D39 for this study is a little puzzling. The authors have noted that D39 is not able to utilise sialic acid, and this is supported by the ManNAc RNA seq data, which suggests that catabolism of sialic acid will be prioritised over galactose. This would indicate that the data generated have limited relevance for aspects of native infection including biofilm formation. There is of course a benefit to using well-defined laboratory strains, but given that the authors were specifically investigating temporal activation of galactose pathways, this would appear to be the incorrect experimental system for this work.

A1. We addressed the reviewer's point about the choice of the strain used in this study both experimentally, and by discussing the available literature in the revised manuscript. As the reviewer rightly mentioned, we wanted to use a well-defined reference strain for which we have had generated historically-relevant data. To address the reviewer's point, we repeated selected experiments using another pneumococcal strain. We found that our results are reproducible with TIGR4 strain that has the ability to utilise sialic acid as a carbon source. Specifically:

- To address the reviewer's point, we have done both growth and RT-PCR experiments. Using a strain with an intact neuraminidase gene (serotype 4 TIGR4 strain), we showed that similar to ManNAc, sialic acid induced metabolic switch away from galactose catabolic pathways as determined by the growth studies (SFigure 1) and RT-PCR (In. 213-223). The relevant text has been added to the modified manuscript as follows: 'To determine whether we could reproduce our results with a strain that has an intact neuraminidase gene, we used serotype 4 TIGR4 strain. When we added 3.5 mM sialic acid on galactose grown pneumococci, a substrate-induced metabolic switch could also be demonstrated with the TIGR4 strain (SFigure 1). Moreover, the addition of either 3.5 mM ManNAc or sialic acid to the pneumococcal cultures grown on galactose led to a significant decrease in the expression of *galK* (3.4 ± 0.07 and 6.6 ± 0.03 , $n=4$ for ManNAc and sialic acid, respectively) and *lacD* (4.5 ± 0.15 and 2.8 ± 0.25 , $n=4$ for ManNAc and sialic acid, respectively) compared to the growth on galactose. Therefore, our observation is reproducible and not strain-specific.'

- The importance of strain and carbon source specific aspects of pneumococcal phenotypes have been discussed in detail (ln.405-415).
- Contrary to the reviewer's claim, in pneumonia model, due to its short assay period, 32 hours, biofilm formation is highly unlikely.

These results show that our data is reproducible, not strain specific, and data generated with the D39 strain is relevant for the aspects of native infection for the hypothesis we aimed to test.

Q2. The interplay between galactose catabolism and biofilm formation is well established in the literature but is largely not supported by the data generated in this study. Despite the authors stating that capsule biosynthesis relies on precursors from galactose catabolism, the $\Delta galK\Delta lacA$ strain produced more biofilm in vitro than the WT. These experiments were also conducted in BHI, which almost guarantees that glucose was being used as the carbon source, subverting the need for galactose utilisation. The in vitro data was also in direct contrast to the in vivo glucuronic acid assay. A Δcps strain should be included both in vitro and in vivo to confirm that these experimental systems are accurately reporting on capsule abundance.

A2. The available literature suggest that galactose increases biofilm formation (Infect Immun. 2016 Oct; 84(10): 2922–2932). In this study, we could not do our experiment solely on galactose because our mutants do not grow on galactose as the sole carbon source. Our biofilm data in BHI shows the complexity of pneumococcal biofilm formation and galactose catabolic pathways' role.

The reviewer states that 'the in vitro data was also in direct contrast to the in vivo glucuronic acid assay'. We believe that the reviewer means RT-PCR data versus glucuronic assay, both used in vivo recovered bacteria (Figure 6). We did not assay glucuronic acid *in vitro*. Using *in vivo* recovered pneumococci mutated in galactose catabolic pathway genes, we detected a significant downregulation of capsule locus gene, *cps2A*, relative to the wild type. Moreover, although the difference was not significant, we could see a trend in the level of glucuronic acid in the mutants compared to the wild type. Therefore, given these data we believe our results from different assays are not contradictory. We explain the reason for this difference to be likely 'due to the differences in the length of time required for both transcription as well as posttranslational control (ln. 338-339)

Unfortunately, we could not recover unencapsulated *S. pneumoniae in vivo* because it is cleared too rapidly. However, we have determined glucuronic acid levels in the wild type ($64 \pm 0.3 \mu\text{M}/108 \text{ CFU}$, $n=4$) and Δcps ($39 \pm 0.1 \mu\text{M}/108 \text{ CFU}$, $n=4$), which had significantly less capsule synthesis compared to the wild type ($p < 0.05$) (ln. 333-334). This supports the conclusion that our assay accurately measuring differences in capsule level.

Q3. The in vivo qRT-PCR data was another source of inconsistency and appeared to be under explored. For example, what were the recovered CFUs for the mutant strains compared to the WT? Are there differences in virulence/host clearance that manifest within the first 4 hours of infection? This could be particularly interesting for the $\Delta lacD$ strain, which is relying on the Leloir pathway at much earlier timepoints.

A3. We did not detect any significant difference in the numbers of recovered bacteria among the pneumococcal strains. The mean recovered CFU/ml for different strains was similar: D39: 1.81×10^7 , $\Delta galK$: 1.9×10^7 , $\Delta lacD$: 1.7×10^7 , and $\Delta galK\Delta lacD$: 1.76×10^7 . This information has been included in the Figure 6 legend.

Q4. There were also some results that did not appear to match the narrative, such as the down-regulation of *pflA* and *pflB*. If the bacterium was indeed using galactose as the primary carbon source, then the genes required for mixed acid fermentation should have been up-regulated in the single mutants (given the suggested redundancy in the Leloir and tagatose pathways). This may suggest that homolactic fermentation was still being primarily utilised and therefore confounding the findings and conclusions regarding galactose utilisation.

A5. The reviewer's statements about *pflA* and *pflB*'s role in mixed acid fermentation is correct, however we interpret these data much differently. In this work, we did not measure the mixed acid fermentation products but the expression of *pflA* and *pflB* was assayed by using the relative quantification method. Our RT-PCR data is showing that relative to the wild type the expression of these genes decreased. This is consistent with what is expected from a mutant strain that has only one or none of the galactose catabolic pathways, which generate substrate, pyruvate, for PflB. Hence, due to attenuated pyruvate generation in the mutants, decreased expression of *pflA* and *pflB* relative to the wild type is expected. We emphasise that decreased relative level of *pflA* and *pflB* does not indicate they are maintaining homolactic fermentation. We included a new sentence to state our interpretation as follows (ln.309-313):

'Decreased expression of *pflA* and *pflB* is consistent with the attenuated galactose metabolism, hence its impact on the decreased synthesis of pyruvate, which is used as a substrate for mixed acid fermentation. It should be noted that PflB requires post-translational activation by PflA. Hence, mRNA levels might not be reflective of the actual PflB activity.'

Q5. The rationale for investigating the *rgg144* and *rgg1518* systems is also not well defined and very little discussion of how these systems intersect with galactose catabolism is offered past reference to earlier works. The impact of removing *rgg144* on *lacA* expression in vitro is quite striking, however this is not replicated in vivo. While this is not unusual for metabolic pathways due to innate complexities in regulation, this finding is not further investigated, or even discussed, and thus has no bearing on the conclusions of the study overall. This appears to be a missed opportunity given the expertise of the authors in this research area.

A5. We added additional background data to define why we studied these systems as suggested by the reviewer:

'In the absence of these systems the pneumococcus is attenuated in colonization and virulence very likely due to their effect on galactose utilization and capsule synthesis. We hypothesized that their impact on galactose utilization may be due to their impact on galactose catabolic pathways'. (ln.315-319).

In addition, we discussed our results in more detail as follows (ln. 417-431):

'The absence of *galK* and *lacD* decreased the expression of *rgg144* RNA, but *rgg1518* RNA expression decreased only in the absence of the tagatose pathway. Differential expression of *rgg144* and *rgg1518* in the galactose catabolic pathway mutants may show that *rgg144* and *rgg1518* are induced through the metabolites generated by the galactose catabolic pathways. Moreover, distinct expression patterns show that despite their conserved structures^{28,37}, the induction of each *rgg* requires a specific stimulus, hence their unique role for the pneumococcus niche-based growth and invasiveness. Given that *rgg144* is part of the core genome and *rgg1518* is found only in 38% of analyzed pneumococcal strains²⁶, we speculate that the Leloir pathway emerged before the tagatose pathway in the evolutionary time frame. When the induction of *PlacA* and *PgalK* were tested in Δ *rgg144* and Δ *rgg1518* backgrounds *in vivo*, it was found that *rgg144* and *rgg1518* induce the

tagatose pathway, but *rgg144* decrease the Leloir pathway. RT-PCR and IVIS results show that there is no bidirectional relationship between *rgg144* and the Leloir pathway expression, probably because of the complexity of regulation *in vivo*'.

Reviewer #1 (Remarks to the Author):

The authors have fully addressed my suggestions, through additional experimentation and clarifications in the text. I believe the work, as presented, represents an important contribution to the field and I congratulate the authors on their efforts.

Reviewer #2 (Remarks to the Author):

The authors have been highly responsive to have satisfactorily addressed all of my concerns.

Reviewer #3 (Remarks to the Author):

The authors have made a reasonable effort to address the reviewer's comments.

The only lingering concern is the biofilm assay data.

I note that all three reviewers flagged this as an issue and it was not sufficiently resolved in the resubmission. The finding that the double mutant produces more biofilm when presumably using glucose as the primary carbon source is interesting. However, this finding was not interrogated to any extent, nor was the underlying mechanism speculated upon. These data would ideally be removed from the manuscript as it appears superfluous and is insufficiently explained.

To ensure compliance with the Editorial Policy Checklist, all graphs should be edited to ensure individual datapoints are presented for all data.

Author response: We are grateful for all reviewers for taking their time to review our manuscript. With their suggestions, the manuscript has improved. Below we addressed third reviewer's only remaining point about our biofilm assay.

Reviewer 3:

Q1. The only lingering concern is the biofilm assay data. i.) I note that all three reviewers flagged this as an issue and it was not sufficiently resolved in the resubmission. ii.) The finding that the double mutant produces more biofilm when presumably using glucose as the primary carbon source is interesting. However, this finding was not interrogated to any extent, nor was the underlying mechanism speculated upon. iii.) These data would ideally be removed from the manuscript as it appears superfluous and is insufficiently explained.

Q2. To ensure compliance with the Editorial Policy Checklist, all graphs should be edited to ensure individual datapoints are presented for all data.

Response to reviewer 3.

Q1i.) I note that all three reviewers flagged this as an issue and it was not sufficiently resolved in the resubmission.

A1i.) We understand the point that the mechanism to explain the biofilm is not fully resolved. Nonetheless, we find the observation very intriguing and thought it would be a valid addition to this manuscript, as it also related to nutritional status. The discussion provides some hypotheses to be explored (ln. 458-473).

Q1ii.) The finding that the double mutant produces more biofilm when presumably using glucose as the primary carbon source is interesting. However, this finding was not interrogated to any extent, nor was the underlying mechanism speculated upon.

A1ii.) In this revised version, we included our explanation of our biofilm results in the Discussion as follows (ln. 458-473):

‘Our data show that in BHI, which contains 27 mM glucose and also complex host glycans, which are rich in non-glucose sugars including galactose, galactose catabolism is acting as a repressor for the biofilm formation. Currently we do not know the exact mechanism of the phenotype observed with the double deletion strain, but it may be due to the imbalance of metabolic precursors important for biofilm formation. It was reported that metabolic regulation plays a central role in the adaptation from the planktonic to biofilm phenotype⁴⁰. Indeed, biofilm forming pneumococci had reduced synthesis of enzymes of the glycolytic pathway and proteins involved in translation, transcription, and virulence, while proteins with a role in pyruvate, carbohydrate, and arginine metabolism increased significantly. Alternatively, in the absence of galactose catabolic pathways in the double mutant, catabolite repression is released, and this may also be important for excess biofilm formation. Overall, our data demonstrates that the processing of carbohydrates plays a key role in the regulation of biofilm development, and further studies are needed to understand the underlying mechanisms for the contribution of galactose catabolic pathways on biofilm synthesis’.

Q1iii.) These data would ideally be removed from the manuscript as it appears superfluous and is insufficiently explained.

A1iii.) We agree that removal of the data would not affect the central message of the paper. However, we would like to keep the data as it is relevant to the nutritional status, and may be of interest to those studying how nutrients influence *S. pneumoniae* phenotypes. We note that reviewers 1 and 2 appear to share this perspective.

Q2. To ensure compliance with the Editorial Policy Checklist, all graphs should be edited to ensure individual datapoints are presented for all data.

A2: This has been done.